# Can eating pleasure be a lever for healthy eating? A systematic scoping review of eating pleasure and its links with dietary behaviors and health

Alexandra Bédard[1], Pierre-Olivier Lamarche[2], Lucie-Maude Grégoire[1,3], Catherine Trudel-Guy[1,3], Véronique Provencher[1,3], Sophie Desroches[1,3], Simone Lemieux[1,3]*

1 Centre Nutrition, santé et société (NUTRISS), Institute of Nutrition and Functional Foods, Université Laval, Québec, QC, Canada, 2 Department of Sociology, Université Laval, Québec, QC, Canada, 3 School of Nutrition, Université Laval, Québec, QC, Canada

* Simone.Lemieux@fsaa.ulaval.ca

**Data Availability Statement:** All relevant data are within the manuscript and its Supporting Information files.

## Abstract

The aims of this review were to map and summarize data currently available about 1) key dimensions of eating pleasure; 2) associations of eating pleasure, and its key dimensions, with dietary and health outcomes and 3) the most promising intervention strategies using eating pleasure to promote healthy eating. Using the scoping review methodology, a comprehensive search of the peer-reviewed literature (Medline, PsycInfo, Embase, ERIC, Web of Science, CINAHL, ABI/Inform global and Sociology Abstract) and of the grey literature (ProQuest Dissertations & Theses and Google) was carried out by two independent reviewers. We included 119 of the 28,908 studies found. In total, 89 sub-dimensions of eating pleasure were grouped into 22 key dimensions. The most frequently found related to sensory experiences (in 50.9% of the documents), social experiences (42.7%), food characteristics besides sensory attributes (27.3%), food preparation process (19.1%), novelty (16.4%), variety (14.5%), mindful eating (13.6%), visceral eating (12.7%), place where food is consumed (11.8%) and memories associated with eating (10.9%). Forty-five studies, mostly cross-sectional (62.2%), have documented links between eating pleasure and dietary and/ or health outcomes. Most studies (57.1%) reported favorable associations between eating pleasure and dietary outcomes. For health outcomes, results were less consistent. The links between eating pleasure and both dietary and health outcomes varied according to the dimensions of eating pleasure studied. Finally, results from 11 independent interventions suggested that strategies focusing on sensory experiences, cooking and/or sharing activities, mindful eating, and positive memories related to healthy food may be most promising. Thus, eating pleasure may be an ally in the promotion of healthy eating. However, systematically developed, evidence-based interventions are needed to better understand how eating pleasure may be a lever for healthy eating.

**Funding:** This work was supported by the Canadian Institutes of Health Research (author who received this grant: Simone Lemieux; grant number FHG 129921; URL: https://cihr-irsc.gc.ca/). There was no additional external funding received for this study. The CIHR had no role in study design, data collection and analysis, decision to publish, or preparation of the manuscript.

**Competing interests:** The authors have declared that no competing interests exist.

## Introduction

Diet quality has been identified as a major determinant of health. However, in developed countries, many individuals still not consume enough fruits, vegetables and wholes grains, while consuming too much processed foods high in energy, sodium, free sugars and saturated fats [1,2]. Despite numerous public health campaigns aimed at improving dietary habits, recent data suggest that diet quality has remained suboptimal over years [3]. These observations are challenging for policymakers and health professionals, underlying the necessity to improve food policies and interventions, and to identify new approaches to promote healthy eating that may have a greater impact on diet quality over time. These efforts could lead to considerable health benefits, along with reducing disease-related costs [4].

Some authors have proposed that the strong focus on the healthiness of food that is typically used to encourage individuals to change their food habits could explain, at least partly, the limited success of strategies used to improve diet quality [5–8]. Indeed, it is likely that the uneven importance attributed to the health value of food compared with other food values (e.g., related to preferences and human experiences) can nurture the beliefs that the healthiness of food is incompatible with these other food attributes [9]. A good example of this is the well-documented unhealthy equal tasty association. Some studies have shown that food presented as being healthy is often considered as being less tasty than "unhealthy" food [10–15]. This perception seems to be influenced by the culture. Previous research observed that, for Americans, unhealthy food is implicitly associated to tastiness [15]. However, the opposite intuition has been observed in France, French people spontaneously associating unhealthy food with bad taste, while linking healthy food to tastiness [16]. Considering that taste is a predominant determinant of food choices [17,18], the unhealthy equal tasty intuition can clearly thwart the adoption of healthy food choices.

A growing number of authors have suggested that pleasure should be emphasized more in the promotion of healthy eating [6–8,19–21]. This is even reflected in the latest dietary guidelines in some countries, including Canada [22] and Brazil [23], but also in intervention strategies using the "intuitive eating" approach (i.e., an approach that implies eating in response to hunger and satiety cues and rediscovering the pleasures of eating) [24]. This new paradigm is supported by a growing body of research that has reported, using different study designs, favorable associations between eating pleasure and dietary and health outcomes. For example, various cross-sectional studies have linked higher eating pleasure to better nutritional status [25], healthier food choices [26–28], increased subjective diet-related quality of life [29] and decreased depressive symptoms [30]. Other studies of both short-term and long-term interventions using eating pleasure also observed promising results. For example, Petit et al. [7] found that focusing on the healthiness of food decreased activity in the gustatory and reward brain areas, whereas focusing on the tastiness of food increased brain activity in self-control, gustatory and reward areas, and most importantly increased the percentage of healthy food choices. Overall, the authors concluded that promoting the tastiness of healthy food may lead to more successful self-control and healthy dietary behaviors. In a randomized controlled trial conducted by our research group [31], a sensory-based intervention addressing the topics of relationship with food, reliance on hunger and satiety cues, food tasting and eating pleasure showed that women who were allocated to the intervention decreased their level of disinhibition in comparison to the control group. Sasson et al. [32] did an experiment with US students who explored food and nutrition from historical, cultural and culinary perspectives during a study abroad program in Italy. Six months after the end of the program, it was found that students had increased their use of fresh and locally grown foods, improved their cooking skills and were less worried about calories [32]. Authors suggested that focusing on eating pleasure

can nurture lifelong healthy eating habits. However, other studies have observed a negative link between eating pleasure and diet quality [27,33,34] or body-mass index (BMI) [35,36]. In some cross-sectional studies, eating pleasure has been associated with the consumption of convenience food [33], late-night snaking [34], and has been identified as a factor motivating the consumption of sweets [13]. Moreover, results from Cornil & Chandon [37] showed that focusing on sensory pleasure made sated eaters and dieters choose larger portions of hedonic foods, while the opposite was observed for hungry eaters and nondieters. Eating pleasure has also been reported as one of the most important reason for weight gain over years in a sample of women and men living with obesity [35].

The use of different approaches to define eating pleasure may partly explain the inconsistency of these outcomes. Studies aimed at understanding overeating and self-regulation failures [21,38–41] mostly describe proper weight management as incompatible with eating pleasure, which they define as visceral impulses triggered by external cues or emotional urges driven by biological preferences for high-sugar, high-fat and high-sodium food. This way of defining eating pleasure is associated with overeating, loss of control and poor diet quality [21,39]. However, other approaches to define eating pleasure have been proposed in the literature [21,42–44]. For example, Cornil & Chandon [21] contrast short-term visceral pleasure with what they call epicurean eating pleasure, or "the enduring pleasure derived from the aesthetic appreciation of the sensory and symbolic value of food". This definition includes not only food characteristics (i.e., taste and appearance) but also other dimensions of the eating experience such as social eating and cooking. They showed that, unlike visceral pleasure, those scoring high on epicurean pleasure preferred smaller portions and displayed a higher level of well-being [21]. Interestingly, epicurean eating pleasure displays similarities with the dimensions of eating pleasure identified in some qualitative studies [42–44]. These studies underline the multidimensional nature of eating pleasure, suggesting it includes dimensions related to food characteristics, the individual, the environment and the social context. The wide variety of definitions of eating pleasure could thus lead to inconsistencies in assessments of the relationship between eating pleasure and healthy eating. Establishing a common understanding of eating pleasure therefore requires identifying its key dimensions and using these to analyze in more detail how eating pleasure is associated with dietary behavior and health outcomes.

Considering the increased interest within the scientific and clinical communities in the integration of eating pleasure in the promotion of healthy eating, a comprehensive mapping of current knowledge about this topic is essential to inform health professionals and policy-makers whether and how the pleasure of eating can be a lever for healthy eating. This will also help to identify gaps that will guide next research efforts in this area. In this comprehensive scoping review we therefore aimed to (1) identify the key dimensions of eating pleasure; (2) map the data currently available about associations of eating pleasure, and its key dimensions, with dietary behaviors and health outcomes and (3) identify most promising intervention strategies using eating pleasure to promote healthy eating. Scoping review design represents a rigorous methodology that allows comprehensive assessment of emerging evidence, and a first step in research development [45]. This technique aims to map the key concepts, types of evidence available, and gaps in research related to a defined field by systematically searching, selecting, and synthesizing existing knowledge [46,47]. This is particularly relevant to disciplines with emerging evidence, since researchers can incorporate a range of study designs in both published and grey literature, and address questions beyond those related to intervention effectiveness [48]. Accordingly, a scoping review was deemed more appropriate than a systematic review or a meta-analysis in the present context because (1) the objectives address a broad, multidisciplinary topic, (2) this topic has been explored using a range of research designs and

(3) this complex and emerging topic has not been reviewed comprehensively before, thereby preventing the identification of more specific objectives that can be the focus of a systematic review or a meta-analysis [49].

## Methods

Based on the Arksey and O'Malley methodological framework for scoping reviews [47] and enhanced according to the recommendations of Levac et al. aimed at clarifying each stage of the framework [48], six steps were performed: (1) identifying the research question, (2) identifying relevant studies, (3) selecting eligible studies, (4) charting the data, (5) collating, summarizing, and reporting results and (6) consultation. The study protocol has been registered on Open Science Framework (https://osf.io/5p7b8/) [50].

### Step 1: Identifying the research questions and eligibility criteria

**Research questions.** The purpose of this review was to assess whether and how eating pleasure can be a lever for healthy dietary behaviors and health. Accordingly, three research questions were addressed:

Q1: How is eating pleasure conceptualized (i.e., key dimensions) in scientific research and organization/government documents?

Q2: What is the current available scientific evidence about associations of eating pleasure, and its key dimensions, with dietary behaviors and health outcomes?

Q3: What are the most promising strategies using eating pleasure to promote healthy dietary behaviors identified through intervention studies?

In parallel, this review sought to identify gaps that research efforts should address to further explore the place of eating pleasure in the promotion of healthy eating, and to inform health professionals and policymakers about how to use eating pleasure in their practical strategies.

Our scoping review was conducted between February 2018 and August 2019. It was conceptualized (SL and AB) and reviewed by coauthors with expertise in scoping review methodology (VP and SD).

**Population and context.** The population targeted was individuals of at least five years old, with no specific condition that would interfere with eating pleasure or alter the perception of eating pleasure. We decided not to include individuals under the age of five because we were interested in eating pleasure as experienced or reported by the individuals themselves, and children under five may have some trouble reporting explicit, conscious judgements due to their limited cognitive maturity, and especially language development [51]. As a result, eating pleasure is usually measured in infants and young children using observer-based measures or parent-rated measures [51,52]. Intervention strategies may also be very different at these ages, focusing more on play-based approaches and often related to neophobia and food familiarity [53–58]. In addition, as we wanted to identify strategies that could be applied in interventions aiming to prevent nutrition-related chronic diseases (e.g., obesity, type 2 diabetes, cardiovascular diseases), we excluded documents targeting people with conditions requiring a strict diet or increased dietary intake. Finally, since issues related to healthy eating differ between underdeveloped and developed countries [59], only studies conducted in developed countries (i.e., countries with high and very high human development index according to the Human Development Report 2016 [60]) were considered for inclusion.

**Concepts.** No *a priori* definition for *eating pleasure* was used since we aimed to conceptualize eating pleasure by identifying its key dimensions in this review. *Scientific research*

referred to peer-reviewed literature and theses/dissertations. Given that this is an emerging area in the scientific literature a broad range of articles and study designs were considered, including but not limited to qualitative, cross-sectional, retrospective, prospective, intervention and mixed methods studies. *Organization/government documents* referred to guidelines, health-promotion tools, reports, and program descriptions published by a health-related organization (i.e., an organization whose ultimate mission is to promote and/or foster the health of individuals/populations) or government. *Dietary behavior* was defined as any actions associated with the act of eating. *Health* was defined according to the definition of the World Health Organization ("a state of complete physical, mental and social well-being and not merely the absence of disease or infirmity" [61]). A *promising strategy* was defined as an intervention method that led to a significant and beneficial impact on dietary behavior and/or health outcomes.

**Inclusion and exclusion criteria.** In addition to responding to our research questions, documents considered for inclusion were: 1) mainly focused on eating pleasure dimensions or reporting data about the links between eating pleasure and dietary behavior and/or health outcomes; 2) targeting individuals of at least five years old; 3) written in English or in French (languages spoken by the research team); 4) conducted in developed countries and 5) interested in eating pleasure as experienced or reported by the individuals themselves. Excluded sources were those that 1) reported results in animals, 2) targeted individuals with a specific condition that would require a strict diet or an increased dietary intake (e.g., type 1 diabetes, malnutrition, decreased appetite, neophobia), 3) targeted people with a condition that may interfere with eating pleasure or alter the perception of eating pleasure (e.g., depression, eating disorder pathologies, pain), 4) reported research objectives aiming to evaluate eating pleasure in relation to a specific food/food group (except for vegetables and fruits, as they are considered as the cornerstone of a healthy diet [62]) and 5) described eating pleasure only in terms of the biological reward mechanisms (e.g., with brain imaging). Conference abstracts, proceedings of symposiums or conferences, editorials, expert opinions, blogs, books, book chapters, book reviews and literature reviews were also excluded.

## Step 2: Identifying relevant studies

With the collaboration of a librarian specializing in nutrition, we developed a comprehensive search strategy using both specific keywords and controlled vocabulary (i.e., standardized terms used by database indexers to describe and categorize articles based on content) related to eating and pleasure. Using May 6, 2018, as a cut-off date, we searched eight electronic databases of peer-reviewed literature: Medline, PsycInfo, Embase, ERIC, Web of Science, CINAHL, ABI/Inform global and Sociology Abstract. This range of databases allowed us to capture a comprehensive sample from the many disciplines related to eating pleasure, namely medical sciences, nutrition, psychology, business, public health, education and sociology. Search terms were adapted for each database and combined using Boolean operators to narrow the results. No date restrictions were implemented in the search. As an example, the Medline search strategy is presented in **Table 1**. Strategies for other databases are available in **S1 Table**. The references of included studies were also scanned to identify other relevant results. To conceptualize eating pleasure in a more comprehensive way and to retrieve unpublished studies, we conducted additional searches in the grey literature using ProQuest Dissertations & Theses Global (cut-off date: July 10, 2019) and the search engine "Google" (between July 15, 2019 and August 26, 2019). For Google, simplified search strategies in English and French (when relevant) were developed and validated by the specialized librarian for organization websites (site:. org) and for the government website of each country for which a relevant peer-reviewed article

**Table 1. The Medline search strategy.**

| Database: Medline |
|---|
| 1. Pleasure/ |
| 2. (pleasur* or pleasant* or fun or enjoy* or epicur* or hedon* or eudaimon* or eudaemon* or eudemon*).tw. |
| 3. 1 or 2 |
| 4. exp FOOD/ |
| 5. (eat* or diet or diets or meal* or food or foods or nutrition).tw. |
| 6. Feeding Behavior/ |
| 7. Food Preferences/ |
| 8. Meals/ |
| 9. DIET/ |
| 10. "DIET, FOOD, AND NUTRITION"/ |
| 11. EATING/ |
| 12. Nutrition Policy/ |
| 13. 4 or 5 or 6 or 7 or 8 or 9 or 10 or 11 or 12 |
| 14. (3 and 13) not (Animals/ not Humans/) |
| 15. Limit 14 to (English or French) |

'exp' indicates that a subject heading is 'exploded' to include all of the narrower subject headings beneath it in the hierarchy.

'/ ' indicates that a term is a subject heading (i.e., controlled vocabulary). Specific keywords are indicated in lines 2 and 5.

'*' represents truncation. For example, "pleasur*" finds terms that begin with the root term "pleasur", such as pleasure, pleasures, pleasurable, etc. '.tw.' is an alias for all of the fields in a database which contain text words and which are appropriate for a subject search.

had been identified in peer-reviewed databases (total of 16 government websites; see S1 Table for search strategies). For Google search strategies, the government websites of the United States, France and Canada as well as organization websites were initially searched. For each of these, the first 100 hits were analyzed. Since the Google engine displays results by relative importance using a link analysis algorithm [63] and all relevant websites were in the first Google hits for these first Google searches, for the remaining governments, we decided that if at least one website was identified as relevant in the first group of 20, then the next 20 websites were searched, for up to a maximum of 100 per government search strategy. Finally, suggestions from authors of this scoping review were gathered to identify additional potentially relevant documents from the literature.

## Step 3: Selecting eligible studies

All retrieved records were imported into EndNote software (Version X9), and duplicate records were deleted. First, two reviewers (POL and LMG) and the research assistant (AB) tested the selection process with 250 articles to ensure a shared understanding of inclusion and exclusion criteria. Thereafter, the two reviewers independently determined the eligibility of all articles using a two-stage screening process consisting of a title and abstract scan followed by a full-text review. If we were unable to obtain the full text of an article, the research assistant contacted the corresponding author for the article in question. For the Google searches, two reviewers (POL and CTG) determined eligibility after reading each web page in detail. Any discrepancies between the two reviewers were resolved through weekly discussion with the research assistant and, if needed, the principal investigator (SL).

## Step 4: Charting the data

A standardized data extraction spreadsheet (Microsoft Excel) was developed by the authors. The following information was extracted: document details, design and methodology, relevant results and authors' conclusions. Subdomains are outlined in **Table 2**. A similar but simplified template was used for web pages. The extraction of data was undertaken by two independent coders (POL and CTG).

Prior to data extraction, this form was pilot tested by the two coders with ten selected records that were sufficiently different in terms of design and objectives to make sure that the

**Table 2. Data charting domains and subdomains.**

| Domain | Subdomain |
|---|---|
| **Document details** | Authors' names |
| | First author's discipline (based on the first author's affiliation) [a] |
| | Name of the peer-reviewed journal [b] |
| | Journal discipline (*Journal citation Reports)* [b] |
| | Title |
| | Year of publication |
| | Language |
| | Country where work was done |
| | Publication type/design |
| | Target population |
| | Words used to designate "pleasure" |
| **Study design and methodology** | Study objective(s) [a] |
| | Participants (number, sex, age, BMI) [a] |
| | Dietary outcomes [a] |
| | Health outcomes [a] |
| | Details about the intervention, if any [a] |
| | • Objective(s) of the intervention |
| | • Theoretical basis of the intervention |
| | • Description of the intervention |
| | • Details on strategies using eating pleasure |
| | • Compensation |
| | • Drop-out rate |
| | Control group(s) [a] |
| | • Description of intervention(s) |
| | • Compensation |
| | • Drop-out rate |
| | Statistical analysis [a] |
| **Relevant results** | Key dimensions of eating pleasure (NVivo, Version 10) |
| | Links between eating pleasure and dietary outcomes [a] |
| | Links between eating pleasure and health outcomes [a] |
| | Promising method(s) for using eating pleasure [a] |
| | Other relevant findings related to eating pleasure [a] |
| **Conclusion [a]** | Conclusion of authors |
| | Source(s) of funding |
| | Contact of the corresponding author |
| | Relevant reference(s) |

[a] Data only extracted for peer-reviewed articles, dissertations and theses.
[b] Data only extracted for peer-reviewed articles.

template would be appropriate for different types of publications. The form was revised as needed during the data extraction to extract relevant information as comprehensively as possible. Each of the two coders extracted data for half of the records. The other coder then revised the extracted data and any discrepancies between the two coders were resolved by discussion and consensus. If consensus was not reached, the research assistant was consulted to resolve the disagreement.

## Step 5: Collating, summarizing and reporting results

As recommended by Levac *et al.* [48], this step was broken into three distinct tasks: (1) analyzing data (including descriptive numerical summary analysis); (2) reporting results and outcomes relevant to the research questions and (3) considering the meaning and implications of the findings for future research, practice and policy. First, a summary of each included document is presented, including the following information: author(s), year of publication, country, publication type, study/document design, journal discipline, first author's discipline, target population, word(s) used to designate "pleasure", and objective(s) targeted by the document. For each research question, a quantitative analysis for each relevant characteristic is presented. Second, data outcomes were analyzed and reported according to research questions. For the first research question (Q1: i.e., key dimensions of eating pleasure), documents analyzed were mainly those focusing on the identification of the dimensions of eating pleasure as well as those reporting data about the links between eating pleasure and dietary behavior and/or health outcomes. Each document was uploaded into qualitative NVivo Software (version 10, QSR International, Burlington, MA) and a thematic analysis of the dimensions of eating pleasure was undertaken. A basic tree node structure was built independently by the two coders and the research assistant based on 10 key articles reporting a detailed approach of eating pleasure [8,42–44,64–69]. These basic tree nodes were then pooled together, and each dimension and sub-dimension of eating pleasure was described based on existing definitions and the literature. The two coders then reviewed all documents independently and extracted all dimensions of eating pleasure. New dimensions and sub-dimensions were created as needed during the extraction process. For peer-reviewed articles and theses, only the methodology and results sections were coded in order not to duplicate findings. The two coders met regularly to compare coding and to discuss all new added nodes and those for which agreement was not perfect (kappa coefficient under 100%). Any discrepancies were discussed and resolved with the research assistant. The tree node structure was discussed with the principal investigator at the beginning, middle and end of the extraction process. For the second research question (Q2: i.e., links between eating pleasure and dietary behaviors and health outcomes), the reported outcome was categorized in line with study authors' judgement as favorable (i.e., eating pleasure was associated with favorable outcomes), unfavorable (i.e., eating pleasure was associated with unfavorable outcomes), mixed (i.e., eating pleasure was associated with both favorable and unfavorable outcomes) or neutral (i.e., eating pleasure was not significantly associated with any outcomes). Outcomes most frequently studied in association with eating pleasure (i.e., diet quality, food choices, portion size, restrained eating, weight/BMI and depressive symptoms) were analyzed separately. Finally, associations between eating pleasure and outcomes were analyzed according to the ways to conceptualize (i.e., dimensions) eating pleasure as identified by coders. For the third research question (Q3: i.e., most promising intervention strategies using eating pleasure to promote healthy eating), we included studies evaluating the effect of an intervention using eating pleasure and also studies that showed any increase in eating pleasure in response to their intervention. For these studies, we made sure to give enough information about the nature of intervention groups, data analysis and measurement of

outcomes in the text and tables in order to fully understand the context in which these studies were carried out. Finally, a table that groups together the dimensions based on their likely favorable or unfavorable impact and ordering the dimensions based on level of evidence was used to summarize the findings. Melnyk & Fineout-Overholt's guidelines were used to assess the level of evidence for each dimension [70]. In this model, evidence is assigned a score from 1 to 7, with 1 being the highest level of evidence. The meaning and implications of the findings for research, practice, and policy, are addressed in the Discussion, along with gaps identified through this review.

### Step 6: Consulting with an expert committee

An expert committee was set up at the beginning of the study, including the authors of this study and two research assistants with expertise in scoping review methodology. This expert committee was consulted for their feedback and deeper understanding of the data at each critical step of our scoping review. In addition, to help to put our findings on the dimensions of eating pleasure into perspective and to interpret them in the context of our population, results were compared to those from focus groups addressing perceptions about eating pleasure in the French-Canadian population conducted earlier by our research team [42].

## Results

From the search, 27,974 records were found from the databases and 934 records identified through other sources (hand-searching references, government and organization websites, and expert committee) for a total of 28,908 records imported into Endnote (**Fig 1**). After exclusion of duplicates, 18,002 records were screened, from which 16,606 were excluded on the basis of Titles/Abstracts. Therefore, 1,396 were reviewed in full and 119 met our eligibility criteria and were included in this review. **S2 Table** provides the references and characteristics of eligible documents. Some documents addressed more than one of our research questions. Of the 119 records, 110 (92.4%) were used for the identification of dimensions of eating pleasure (Q1), 45 (37.8%) documented the links between eating pleasure and dietary behaviors and health (Q2), and 12 (10.1%) intervention studies were used to identify most promising methods using eating pleasure to promote healthy dietary behaviors (Q3).

### Q1—Key dimensions of eating pleasure

**Description of included documents.** Of the 110 documents that identified dimensions of eating pleasure, 66 (60.0%) were peer-reviewed articles identified through scientific databases, five (4.5%) were theses and dissertations and 39 (35.5%) were governmental or organizational web pages (**Table 3** and S2 Table). The first author's discipline of peer-reviewed articles as well as of theses and dissertations (n = 71) was most often psychology (n = 18; 25.4%), followed by business, economy and management (n = 16; 22.5%), nutrition (n = 8; 11.3%) and food sciences (n = 7; 9.9%). For peer-reviewed articles (n = 66), one third (n = 26; 39.4%) were published in a journal for which the major discipline was psychology, followed by nutrition (n = 9; 13.6%), public health (n = 8; 12.1%), food sciences (n = 7; 10.6%), business, economy and management (n = 6; 9.1%). For governmental and health-related organizational web pages (n = 39), 13 (33.3%) were dietary guidelines, 11 (28.2%) were healthy eating promotion tools, nine (23.1%) were reports and six (15.4%) were program descriptions. In total, 88 documents (80.0%) were written in English and 22 (20.0%) in French. Most of the documents were published after 2000 (93.6%), and 70.9% after 2010. Most represented countries were France (n = 28; 25.5%), United States of America (n = 24; 21.8%), Canada (n = 16; 14.5%), Australia (n = 10; 9.1%), Finland (n = 8; 7.3%) and United Kingdom (n = 6; 5.5%). The words most

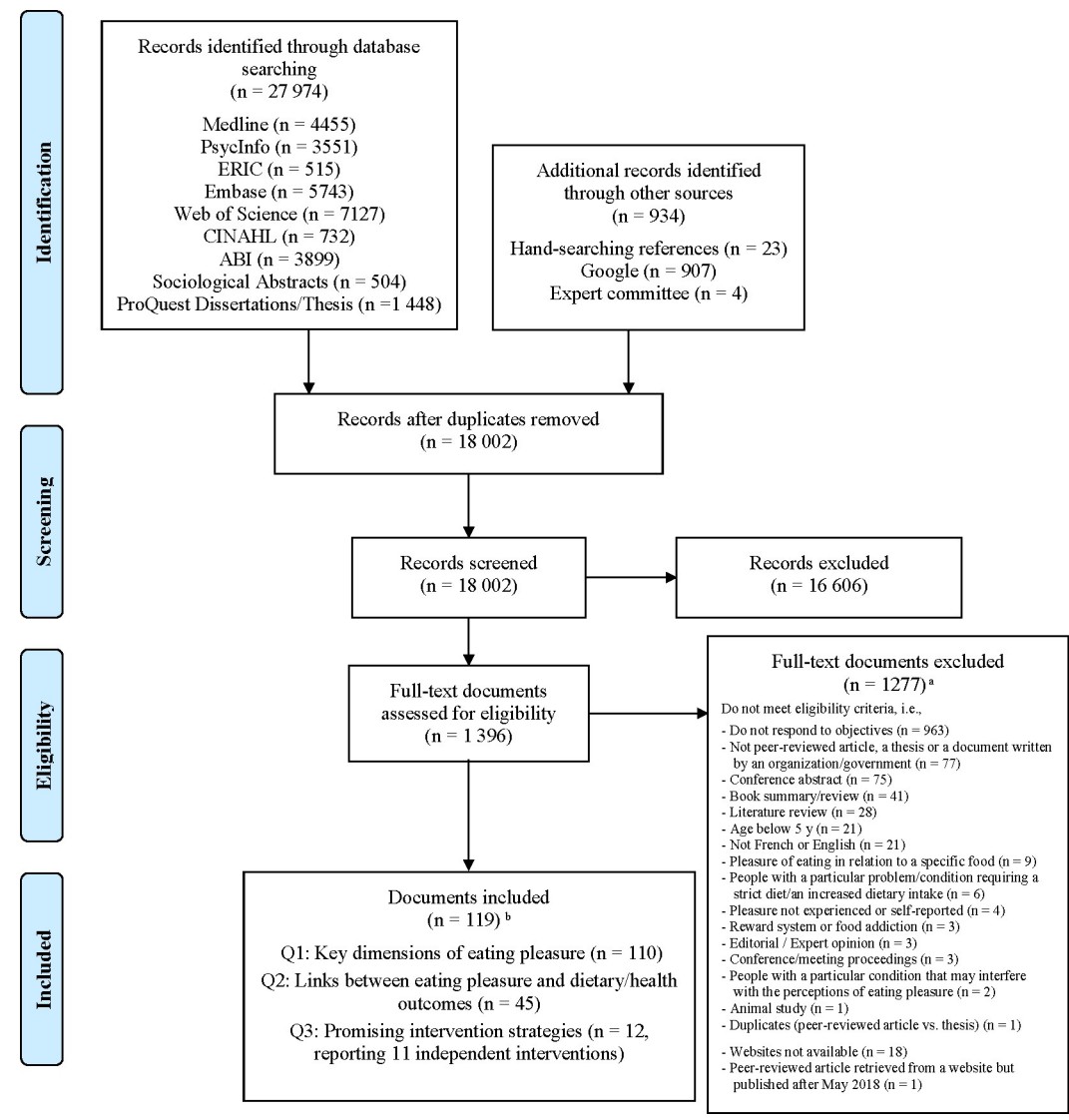

**Fig 1. Flow chart of the scoping review process.** [a] For the exclusion criteria, the reason identified for each document is the one that the coders identified first. This does not exclude the possibility that other criteria were not met. [b] Some documents addressed more than one research question.

often found to designate "pleasure" were "pleasur*" or "plaisir" (French for pleasure) (n = 58; 52.7%), followed by "enjoy*" (n = 26; 23.6%), "hedon*" (n = 5; 4.5%), "fun" (n = 3; 2.7%), "happiness" (n = 1; 0.9%) or a mix of the above words (n = 17; 15.5%). The populations targeted were young and middle-aged adults in half of the documents (n = 56; 50.9%), followed by children (n = 9; 8.2%), older adults (n = 6; 5.5%), adolescents (n = 5; 4.5%), children and adolescents (n = 4; 3.6%), children, adolescents and young and middle-aged adults (n = 3; 2.7%), children and young and middle-aged adults (n = 2; 1.8%), and adolescents and young and middle-aged adults (n = 1; 0.9%). In addition, 24 (21.8%) documents targeted all population strata.

**Key dimensions of eating pleasure.** The key dimensions of eating pleasure are presented in Table 4. The dimensions were identified by two independent coders, quoting phrases in documents that could complete the sentence "The pleasure of eating is. . .". Eighty-nine eating

**Table 3. General characteristics of included documents.**

| | Q1 [a] (n = 110) n (%) | Q2 [a] (n = 45) n (%) | Q3 [a] (n = 11[b]) n (%) |
|---|---|---|---|
| **Publication type** | | | |
| Peer-reviewed article | 66 (60.0) | 44 (97.8) | 11 (100) |
| Thesis/Dissertation | 5 (4.5) | 1 (2.2) | 0 |
| Web page | 39 (35.5) | N/A | N/A |
| **Study/document design** | | | |
| *Peer-reviewed articles / Theses* | | | |
| Cross-sectional study | 31 (28.2) | 28 (62.2) | 0 |
| Qualitative study | 17 (15.5) | 2 (4.4) | 0 |
| Intervention study–Single exposure [c] | 16 (14.5) | 8 (17.8) | 7 (63.6) |
| Mixed-design study | 5 (4.5) | 1 (2.2) | 0 |
| Intervention study–Prolonged exposure [c] | 1 (0.9) | 5 (11.1) | 4 (36.4) |
| Prospective study | 1 (0.9) | 1 (2.2) | 0 |
| *Web pages* | | | |
| Guidelines | 13 (11.8) | N/A | N/A |
| Healthy eating promotion tool | 11 (10.0) | N/A | N/A |
| Report [d] | 9 (8.2) | N/A | N/A |
| Program description [e] | 6 (5.5) | N/A | N/A |
| **Language** | | | |
| English | 88 (80.0) | 45 (100) | 11 (100) |
| French | 22 (20.0) | 0 | 0 |
| **Year of publication** | | | |
| Before 2000 | 7 (6.4) | 3 (6.7) | 0 |
| 2000–2004 | 11 (10.0) | 4 (8.9) | 0 |
| 2005–2009 | 14 (12.7) | 7 (15.6) | 1 (9.1) |
| 2010–2014 | 34 (30.9) | 11 (24.4) | 4 (36.4) |
| 2015–2019 | 44 (40.0) | 20 (44.4) | 6 (54.5) |
| **Country** | | | |
| France | 28 (25.5) | 5 (11.1) | 1 (9.1) |
| United States of America | 24 (21.8) | 13 (28.9) | 3 (27.3) |
| Canada | 16 (14.5) | 5 (11.1) | 1 (9.1) |
| Australia | 10 (9.1) | 2 (4.4) | 0 |
| Finland | 8 (7.3) | 3 (6.7) | 0 |
| United Kingdom | 6 (5.5) | 6 (13.3) | 3 (27.3) |
| Denmark | 2 (1.8) | 0 | 0 |
| Germany | 2 (1.8) | 1 (2.2) | 1 (9.1) |
| Japan | 2 (1.8) | 1 (2.2) | 0 |
| Brazil | 1 (0.9) | 0 | 0 |
| Greece | 1 (0.9) | 1 (2.2) | 0 |
| Ireland | 1 (0.9) | 0 | 0 |
| Italy | 1 (0.9) | 1 (2.2) | 0 |
| Malaysia | 1 (0.9) | 2 (4.4) | 0 |
| Netherlands | 1 (0.9) | 0 | 0 |
| Taiwan | 1 (0.9) | 1 (2.2) | 0 |
| Vietnam | 1 (0.9) | 1 (2.2) | 0 |

(*Continued*)

**Table 3.** (*Continued*)

| | Q1 [a]<br>(n = 110)<br>n (%) | Q2 [a]<br>(n = 45)<br>n (%) | Q3 [a]<br>(n = 11[b])<br>n (%) |
|---|---|---|---|
| Multi-country | 4 (3.6) | 3 (6.7) | 2 (18.2) |
| **Words used to designate "pleasure" [f]** | | | |
| Pleasur* OR Plaisir | 58 (52.7) | 21 (46.7) | 5 (45.5) |
| Enjoy* | 26 (23.6) | 15 (33.3) | 5 (45.5) |
| Hedon* | 5 (4.5) | 1 (2.2) | 0 |
| Fun | 3 (2.7) | 0 | 0 |
| Happiness | 1 (0.9) | 0 | 0 |
| More than one of the above words | 17 (15.5) | 8 (17.8) | 1 (9.1) |
| **Journal discipline [g]** | | | |
| Psychology | 26 (39.4) | 16 (36.4) | 3 (27.3) |
| Nutrition | 9 (13.6) | 10 (22.7) | 5 (45.5) |
| Public Health | 8 (12.1) | 4 (9.1) | 1 (9.1) |
| Food sciences | 7 (10.6) | 5 (11.4) | 0 |
| Business, Economy & Management | 6 (9.1) | 2 (4.5) | 1 (9.1) |
| Multidisciplinary sciences | 3 (4.5) | 2 (4.5) | 1 (9.1) |
| Social sciences | 3 (4.5) | 0 | 0 |
| Medicine | 2 (3.0) | 3 (6.8) | 0 |
| Recreation | 2 (3.0) | 2 (4.5) | 0 |
| **First author's discipline [h]** | | | |
| Psychology | 18 (25.4) | 13 (28.9) | 4 (36.4) |
| Business, Economy & Management | 16 (22.5) | 7 (15.6) | 2 (18.2) |
| Nutrition | 8 (11.3) | 8 (17.8) | 3 (27.3) |
| Food sciences | 7 (9.9) | 4 (8.9) | 0 |
| Social sciences | 4 (5.6) | 1 (2.2) | 0 |
| Medicine | 3 (4.2) | 4 (8.9) | 2 (18.2) |
| Public Health | 3 (4.2) | 3 (6.7) | 0 |
| Recreation | 2 (2.8) | 1 (2.2) | 0 |
| Communication | 1 (1.4) | 0 | 0 |
| Patient-centred outcomes | 1 (1.4) | 0 | 0 |
| Physical activity | 1 (1.4) | 1 (2.2) | 0 |
| Rehabilitation | 1 (1.4) | 0 | 0 |
| Multiple fields | 6 (8.5) | 3 (6.7) | 0 |
| **Target population [i]** | | | |
| Young and middle-aged adults | 56 (50.9) | 34 (75.6) | 9 (81.8) |
| Children | 9 (8.2) | 3 (6.7) | 1 (9.1) |
| Older adults | 6 (5.5) | 2 (4.4) | 0 |
| Adolescents | 5 (4.5) | 4 (8.9) | 0 |
| Children and adolescents | 4 (3.6) | 0 | 0 |
| Children, adolescents and young and middle-aged adults | 3 (2.7) | 0 | 0 |
| Children and young and middle-aged adults | 2 (1.8) | 1 (2.2) | 1 (9.1) |
| Adolescents and young and middle-aged adults | 1 (0.9) | 1 (2.2) | 0 |

(*Continued*)

**Table 3.** (Continued)

| | Q1 [a] (n = 110) n (%) | Q2 [a] (n = 45) n (%) | Q3 [a] (n = 11[b]) n (%) |
|---|---|---|---|
| All | 24 (21.8) | 0 | 0 |

[a] Q1: How is eating pleasure conceptualized (i.e., key dimensions) in scientific research and organization/government documents? Q2: What is the current available scientific evidence about associations of eating pleasure, and its key dimensions, with dietary behaviors and health outcomes? Q3: What are the most promising strategies using eating pleasure to promote healthy dietary behaviors identified through intervention studies?

[b] Twelve documents have been identified for the research question 3, from 11 independent interventions. We report in this table the characteristics of the 11 independent interventions.

[c] Single exposure indicates that participants were exposed to a treatment on only one occasion (acute exposure) while prolonged exposure indicates that participants were exposed to treatment(s) on several occasions (i.e., longer-term interventions).

[d] Reports included documents labelled "report" and covering various topics such as market analysis, food trends, survey results and committee recommendations.

[e] Program descriptions included documents detailing objectives and/or strategies of a health-based program in which eating pleasure was integrated.

[f] For articles, theses and dissertations, words identified as designating pleasure were those found in the title/abstract. For web pages, words identified were those found in the entire document.

[g] The journal name was used to find the associated discipline using Journal Citation Reports (InCites, 2018). When the journal was not indexed in this database, the scope of the journal was used to identify the discipline. See S3 Table for the description of each discipline. Theses, dissertations and web pages were excluded from Journal Discipline analysis. Q. 1: n = 66; Q. 2, n = 44 and Q. 3, n = 13.

[h] The first author's affiliation was used to determine the author's discipline. See S3 Table for the description of each discipline. Multiple fields included: Nutrition / Physical activity (n = 2), Business, Economy & Management / Recreation (n = 2), Business, Economy & Management / Physical activity (n = 1) and Social sciences / Nutrition (n = 1). Web pages were excluded from First Author's Discipline analysis. Q. 1: n = 71; Q. 2, n = 45 and Q. 3, n = 13.

[i] For articles, theses and dissertations, the target population was children when the mean age was between 5 and 12 years, adolescents when the mean age was between 12 and 18 years, young and middle-aged adults when the mean age was between 18 and 65 years and older adults when the mean age was over 65 years. When no mean age was provided, the age range was used to identify the target population most represented. For web pages, we used the target population identified in the document by the authors. The target population was considered as "all" when the document targeted all strata of the population or when no target population was identified by authors.

pleasure sub-dimensions were identified through the literature review and grouped into 22 key dimensions (see **S4 Table** for the complete description of each sub-dimension). In order of importance, eating pleasure was related to (1) sensory experiences (n = 56; 50.9%; e.g., taste, appearance, texture), (2) social experiences (n = 47; 42.7%; e.g., eating with others, preparing meals with others, respecting shared norms and practices such as culture and traditions), (3) food characteristics besides sensory attributes (n = 30; 27.3%; e.g., healthy, unhealthy, fresh), (4) food preparation process (n = 21; 19.1%; e.g., cooking, gardening, grocery shopping), (5) novelty (n = 18; 16.4%; e.g., discovering new food, dishes and tastes, learning about food), (6) variety (n = 16; 14.5%; e.g., in type of food, flavors and ways that food is prepared), (7) mindful eating (n = 15; 13.6%), (8) visceral eating (n = 14; 12.7%; e.g., eating to reward yourself, to cope with emotions), (9) place where food is consumed (n = 13; 11.8%; e.g., eating at restaurants, eating while travelling, eating at home), (10) memories associated with eating (n = 12; 10.9%), (11) atmosphere where food is consumed (n = 11; 10.0%), (12) psychological and physical state during food intake (n = 11; 10.0%; e.g., experiencing emotions when eating), (13) food anticipation (n = 10; 9.1%), (14) special occasions (n = 10; 9.1%), (15) having the choice (n = 10; 9.1%), (16) food intake structure (n = 9; 8.2%; e.g., diet structure, balanced diet, meal composition), (17) taking time (n = 9; 8.2%), (18) health considerations (n = 8; 7.3%; e.g., balancing pleasure and health, making healthy choices), (19) eating according to food preferences (n = 8; 7.3%), (20) the psychological and physical state after food intake (n = 7; 6.4%; e.g., feeling full), (21) respecting eating habits (n = 4; 3.6%) and (22) ideological considerations (n = 4; 3.6%; e.g., environmental movement).

**Table 4. Key dimensions of eating pleasure.**

| Dimensions | Number of records (n = 110) n (%) | Description | Sub-dimensions (n) | References |
|---|---|---|---|---|
| Sensory experiences | 56 (50.9) | To experience food / meals with the five senses, i.e., to experience its sensory qualities. | Taste (48); Appearance (18); Texture (14); General sensory aspects (13); Smell (12); Aesthetic (5); Sound (4); Temperature (1) | [7,8,15,21–23,25,26,30,33,37,42–44,65,67,68,71–109] |
| Social experiences | 47 (42.7) | To integrate social experiences into the act of eating. | Eating with others (36); Preparing meals with others (13); Respecting shared norms and practices (Culture & Traditions, Social rituals, Citizen responsibility, Identity claim) (13); General social experiences (7); Receiving and serving people (4); Serving food that our loved ones like (4); Eating alone (3); Knowing the person who produces / prepares the food / meal (3); Letting yourself be served (2) | [22,23,27,29,36,42–44,65,67–69,71,72,75,78,80,81,91,92,95,98–104,106,107,110–126] |
| Food characteristics | 30 (27.3) | To eat food with certain attributes, other than those related to sensory ones. | Healthy (10); Types of food (7); Unhealthy (6); Fresh (4); Convenient / simple (3); Local (3); Natural (3); Satiating (3); Special (3); Well-known (3); Good nutritional value (2); Original (2); Well-prepared (2); Adequate energy content (1); Authentic (1); Organic (1); Homemade (1) | [8,15,16,27,36,42–44,65,67,68,71,75,78,80–82,85,91,92,97,104,119,121,127–132] |
| Food preparation process | 21 (19.1) | To experience the steps involved in buying and preparing food before eating it. | Cooking (19); Gardening (5); Grocery shopping (5); Going to the farmers' market (2) | [22,23,28,42–44,68,72,91,92,100,104,113,118–120,122,133–136] |
| Novelty | 18 (16.4) | To learn something about food or to experiment with something new or unusual in terms of taste, foods or meals. | Discovering new foods, dishes and tastes (14); Learning about food (6); Breaking the routine (2) | [22,27,33,36,42,44,67,68,71,72,89,92,104,107,110,113,122,137] |
| Variety | 16 (14.5) | To consume a diet whose components are varied. | Variety in type of food (9); Variety in flavors (6); Variety in the way food is prepared (5) | [22,23,27,36,42–44,71,96,99,122,124,132,137–139] |
| Mindful eating | 15 (13.6) | To consume food while savoring every bite; being aware of the present moment and not being distracted. | — | [23,43,44,96,97,108,116–118,139–144] |
| Visceral eating | 14 (12.7) | To satisfy short-term visceral impulses triggered by hunger, external cues or internal emotional cues. | Rewarding yourself (10); Coping with emotions (5); Eating impulsively, based on appeal (1) Disinhibition (1); Eating in response to external cues (1) | [8,13,21,34,42–44,68,79,85,92,107,145,146] |
| Place | 13 (11.8) | To consume food in a specific environment or place. | Eating at restaurant (11); Eating while travelling / tourism (5); Eating at home (3); Eating outdoors (2); Eating at work / school (1); Eating in front of the television (1) | [42–44,65,67–69,92,98,99,107,110,126] |

(*Continued*)

**Table 4.** (Continued)

| Dimensions | Number of records (n = 110) n (%) | Description | Sub-dimensions (n) | References |
|---|---|---|---|---|
| Memories | 12 (10.9) | To consider food / meals through memorable food experiences from the past. | — | [42,43,67–69,72,89,112,125,126,147,148] |
| Atmosphere | 11 (10.0) | To benefit from an atmosphere created by lighting, music, ambient temperature and decoration, among other things, while eating a food / meal. | — | [22,23,43,44,67,68,99,107,118,127,142] |
| Psychological / physical state during food intake | 11 (10.0) | To experience a psychological and physical well-being state when eating. | Experiencing emotions (9); Experiencing a physical well-being state in general (2); Experiencing a sense of relaxation (2); Experiencing a sense of satisfaction (2); Experiencing an alliesthesia state (1) | [33,43,80,81,85,90,92,137,149–151] |
| Food anticipation | 10 (9.1) | To anticipate the consumption of a food / meal. | — | [43,69,71,77,92,112,125,126,151,152] |
| Special occasions | 10 (9.1) | To consume food on special occasions (e.g., holidays, birthday parties, weekends) | — | [8,42,65,67,68,71,79,98,144,153] |
| Having the choice | 10 (9.1) | To feel free to choose food or the place where food is eaten | — | [8,42–44,99,110,113,119,121,153] |
| Food intake structure | 9 (8.2) | To structure food intake throughout the day according to one's preferences and in a balanced way. | Diet structure (5); Balanced diet (4); Meal composition (3); Meal structure (1); Portion size (1) | [28,42–44,71,72,78,99,130] |
| Taking time | 9 (8.2) | To take time buying, preparing and consuming food, to eat in a relaxed environment. | — | [42–44,65,68,104,121,137,153] |
| Health consideration | 8 (7.3) | To make food choices and develop eating habits that help maintain good health. | Balancing pleasure and health (6); Making healthy choices (5); Restraining food intake (2) | [8,22,42,71,78,82,90,129] |
| Food preferences | 8 (7.3) | To choose or eat the food one prefers, to respect one's food preferences. | — | [27,36,68,71,81,82,154,155] |
| Psychological / physical state after food intake | 7 (6.4) | To experience a psychological and physical well-being state after eating a meal / snack. | Feeling full (4); Having energy (2); Feeling relax (2); Feeling satisfied (2); Feeling as meeting his/her body needs (2); Feeling light (1) | [42–44,71,75,137,138] |
| Eating habits | 4 (3.6) | To choose or eat certain foods in accordance with eating habits, i.e., to eat food that the person is used to eating. | — | [8,43,65,71] |
| Ideological considerations | 4 (3.6) | To choose and eat food considering the ideological dimensions one values, including its political, intellectual and spiritual dimensions. | Environmental movement (2); General intellectual consideration (2); Personal growth (1) | [44,91,92,104] |

## Q2. Links between eating pleasure and dietary behavior/health outcomes

**Description of included documents.** Details about characteristics of scientific documents (peer-reviewed articles and theses; n = 45) linking eating pleasure to dietary behavior and health outcomes are presented in Tables 3 and 5. Most studies were cross-sectional (n = 28; 62.2%), followed by short-term, single exposure intervention studies (n = 8; 17.8%), longer-term, prolonged exposure intervention studies (n = 5; 11.1%), qualitative studies (n = 2; 4.4%), mixed-design studies (n = 1; 2.2%) and prospective studies (n = 1; 2.2%; Table 3). Almost all documents were published after 2000 (n = 42; 93.3%), and two thirds were published after 2010 (n = 31; 68.9%; Table 3). Studies were carried out in the United States of America (n = 13; 28.9%), United Kingdom (n = 6; 13.3%), France (n = 5; 11.1%), Canada (n = 5; 11.1%), Finland (n = 3; 6.7%), Australia (n = 2; 4.4%), Malaysia (n = 2; 4.4%), Germany (n = 1; 2.2%), Japan (n = 1; 2.2%), Greece (n = 1; 2.2%), Italy (n = 1; 2.2%), Taiwan (n = 1; 2.2%), Vietnam (n = 1; 2.2%) or in multiple countries (n = 3; 6.7%; Table 3). Studies included both men (boys) and women (girls) (n = 36; 80.0%) or only women (girls) (n = 6; 13.3%), and in three studies (6.7%), the sex of participants was not specified (Table 5). In total, 126,368 participant units (67.7% women) were included in the analyses of all studies, with sample size ranging from 12 to 50,003 participants. Most studies included young and middle-aged adult participants only (mean age between 18 and 65 years; n = 34; 75.6%) while fewer included adolescents only (n = 4; 8.9%), children only (n = 3; 6.7%), older adults only (n = 2; 4.4%), children and young and middle-aged adults (n = 1; 2.2%), and adolescents and young and middle-aged adults (n = 1; 2.2%). Most studies did not report participants' BMI (n = 25; 55.6%).

**Measures of eating pleasure.** Tools used in these studies to measure eating pleasure are presented in Table 5. Eating pleasure was measured using a variety of tools, namely interviews (n = 2) [89,137], pairing and categorization tasks (n = 1) [26], single items (n = 11) [28,29,76,109,111,140,141,147,148,152,160], multi-item questionnaires developed by authors for the purpose of the study (n = 8) [21,74,112,136,145,148,150,156], adapted versions of existing multi-item questionnaires (n = 7) [13,33–35,75,157,158] or existing multi-item questionnaires (n = 13) [21,25,27,30,36,74,84,86,88,125,126,146,151]. In total, 37 different tools were used to measure eating pleasure: 1 interview process, 2 pairing and categorization tasks, 15 single items and 19 questionnaires. Questionnaires used to measure eating pleasure were (1) the Pleasures Questionnaire (Appleton & McGowan, 2006) [156], (2) the pleasure subscale of the Health and Taste Attitude Scale (Roininen et al., 1999) [25,84,86,88], (3) an international survey on the causes and treatment of obesity (Bray et al., 1992) [35], (4) the Epicurean Eating Pleasure Questionnaire (Cornil & Chandon, 2016) [21], (5) the external eating and emotional eating subscales of the Dutch Eating Behavior Questionnaire (DEBQ; van Strien et al., 1986) [21], (6) the pleasure subscale of the Dish Choice Questionnaire (Ducrot et al., 2015) [27,36], (7) the enjoyment of food subscale of the Adult's Eating Behaviors Questionnaire (Hunot et al., 2016) [157], (8) an emotions questionnaire (Richins, 1997) [151], (9) the pleasure motivation subscale of the Motivation to Eat Questionnaire (Jackson et al., 2003) [145,146], (10) the pleasure subscale of a questionnaire developed by Lindeman et al. (1999) and assessing food choice motives [74], (11) the pleasure subscale (sensory appeal) of The Food Choice Questionnaire (Steptoe et al. 1995) [30,74,75], (12) the enjoyment of food subscale of the self-reported version of the Children's Eating Behaviors Questionnaire (Loh et al., 2013) [158], (13) the pleasure eating subscale of the Meanings of Eating Index (MEI; McClain et al., 2011) [150], (14) a questionnaire assessing hedonic consumption values (Babin et al. 1994) [33], (15) the pleasure motivation subscale of the Eating Motivation Survey (adapted from Renner et al., 2012) [13,34], (16) the pleasure subscale of a questionnaire assessing consumption attitudes (Remick et al., 2009) [111], (17) the pleasure subscale of a questionnaire developed to explore

Table 5. Characteristics of included studies (n = 45) reporting links between eating pleasure and dietary/health related outcomes.

| Reference | Country | Study aim | Sample | Measurement of eating pleasure | Favorable links | Unfavorable links | Neutral links |
|---|---|---|---|---|---|---|---|
| **Qualitative studies (n = 2)** | | | | | | | |
| Yang et al., 2014 [137] | Malaysia | To explore the perception of Malaysian Chinese towards food and eating. | 7 women, 5 men Age range: 22–34 BMI: N/A | A qualitative research method known as Zaltman Metaphor Elicitation Technique (ZMET) that combines the use of respondents' self-collected images with a series of in-depth interview tools | Brings happiness | — | — |
| Yang & Khoo-Lattimore, 2015 [89] | Taiwan | To explore the food perception of young Taiwanese consumers. | 9 women, 3 men Age range: 25–30 BMI: N/A | A qualitative research method known as Zaltman Metaphor Elicitation Technique (ZMET) that combines the use of respondents' self-collected images with a series of in-depth interview tools | Brings happiness | — | — |
| **Mixed-design studies (n = 1)** | | | | | | | |
| Prior & Limbert, 2012 [111] | UK | To explore adolescents' perceptions and experiences of family meals, to identify factors relating to frequency and importance of family meals and to compare results between males and females. | 48 girls, 28 boys Age range: 13–14 BMI: N/A | The item "Mealtimes are generally enjoyable" | ↑ Importance of family meals (girls) | — | Importance of family meals (boys) Family meal frequency (girls and boys) |
| **Cross-sectional studies (n = 28)** | | | | | | | |
| Ainuki et al., 2013 [29] | Japan | To examine the association of eating experiences in childhood with eating behavior and subjective diet-related quality of life (SDQOL) in adulthood using data from a population-based survey. | 1 592 women, 1 344 men Age range: 20 and + Mean age: 53.9 ± 15.8 BMI: N/A | The item: "I enjoyed family mealtimes and felt comfortable during them." | ↑ balanced diet ↑ vegetable dishes ↑ Subjective diet-related quality of life | — | Dishes cooked with oil (deep fried or stir fried) Skip breakfast |
| Appleton & McGowan, 2006 [156] | UK | To investigate the importance of pleasure normally associated with eating in the relationship between restrained eating and poor psychological health. | 123 students Age range: 18–23 BMI: N/A | The Pleasures Questionnaire, developed specifically for the study, that consists of 10 activities commonly participated in by the study population, in order to bring pleasure: smoking, eating, exercising, drinking alcohol, sexual activity, socialising, listening to music, playing computer games, driving, watching television. Respondents report the amount of pleasure normally associated with participating in each activity. Only responses to the Eating item were used in the study. | — | — | Anxiety state Depressive state Satisfaction with life |
| Bailly et al., 2015 [25] | France | To identify the predictors of nutritional status in order to identify both common and sex specific predictive pathways in an aging population. | 319 women, 145 men Age range: 65–97 Mean age: 77.41 ± 7.48 BMI: N/A | The 5-item Pleasure of eating subscale of the Health and Taste Attitude Questionnaire (HTAQ —Roininen et al., 1999). | Better nutritional status | — | — |

(Continued)

Table 5. (Continued)

| Reference | Country | Study aim | Sample | Measurement of eating pleasure | Favorable links | Unfavorable links | Neutral links |
|---|---|---|---|---|---|---|---|
| Cachelin et al., 1998 [35] | USA | To investigate the beliefs and attitudes toward weight gain of a large non-clinical sample of women and men with obesity. | 1 674 women, 1 720 men. Mean age: 47.0. BMI: obesity. Mean BMI: 36.6 | Participants were asked to rate how important they think the following factors are in explaining why they have gained weight in the past: genetics (biology inherited from parents); slow metabolism; cravings for carbohydrates; depression; stress; lack of willpower; weight cycling (yo-yo dieting); lack of exercise; low self-esteem; need to avoid social or sexual situations; just enjoy eating. These questions were modeled after those used by Bray et al. (1992). | — | Perception from participants that eating pleasure is the reason for their weight gain | — |
| Cornil & Chandon, 2016 [21] | USA | To refine the understanding of the concept of Epicurean eating pleasure and how it differs from visceral eating pleasure. | 152 women, 99 men. Age range: 18 and +. BMI: 56% were overweight or obese | Epicurean eating pleasure was measured using a 7-item questionnaire developed and validated by the authors. Visceral eating pleasure was measured using the 10-item external eating subscale and the 13-item emotional eating subscale of the Dutch Eating Behavior Questionnaire (DEBQ, van Strien et al., 1986). | Preference for smaller food portions (Epicurean); ↑ Wellbeing (Epicurean) | Preference for larger food portions (Visceral); ↓ Wellbeing (Visceral); ↑ Restrained eating (Visceral); ↑ BMI (Visceral) | Restrained eating (Epicurean); BMI (Epicurean); Health worries (Epicurean and Visceral) |
| Crawford et al., 2007 [28] | Australia | To examine associations between shopping, food preparation, meal and eating behaviors and fruit and vegetable intake among women. | 1580 women. Age range: 18–65. BMI: N/A | The three items: "I enjoy food shopping", "I enjoy the evening meal", "How often do you enjoy cooking?". | ↑ intakes of fruits and vegetables | — | — |
| Ducrot et al., 2016 [36] | France | To explore the association between the motives behind dish choices (including pleasure motive) during home meal preparation and overweight (including obesity) in a large sample of French adults. | 39 676 women, 10 327 men. Age range: 18 and +. BMI: 32.6% were overweight or obese | The 5-item pleasure subscale of the Dish Choice Questionnaire (Ducrot et al., 2015) | — | ↑ overweight/obesity | — |
| Ducrot et al., 2017 [27] | France | To evaluate the difference in diet quality according to the importance attached by individuals to various dish choice motives (including pleasure motive) for home-meal preparation and adherence to nutritional guidelines, as well as energy and food intakes. | 26 406 women, 21 604 men. Age range: 18 and +. BMI: N/A | The 5-item pleasure subscale of the Dish Choice Questionnaire (Ducrot et al., 2015) | ↓ Energy intake; ↓ Convenience food consumption | ↓ Adherence to nutritional guidelines | — |
| Hunot et al., 2016 [157] | UK | To adapt the Child Eating Behavior Questionnaire (CEBQ) into a self-report Adult Eating Behavior Questionnaire (AEBQ) to explore whether the associations between appetitive traits and BMI observed in children are present in adults. | 919 women, 743 men. Age range: 18 and +. BMI: 52.7% were overweight or obese | The 3-item Enjoyment of Food subscale of the Adult Eating Behavior Questionnaire (AEBQ), a questionnaire adapted from the child version (Wardle et al., 2001). | — | ↑ BMI | — |

(Continued)

**Table 5.** (Continued)

| Reference | Country | Study aim | Sample | Measurement of eating pleasure | Favorable links | Unfavorable links | Neutral links |
|---|---|---|---|---|---|---|---|
| Hur & Jang, 2015 [151] | USA | To explore the role of affective responses, particularly anticipated emotions, in consumers' decision-making behaviors considering the unique aspects of a healthy food consumption context. | 406 women, 403 men<br>Age range: 18 and +<br>Mean age: 33.2<br>BMI: N/A | To measure anticipated pleasure, respondents were asked to imagine a hypothetical situation (e.g., "If I ate the advertised menu items, I would feel...) and rated their anticipated feelings of "pleasure, happiness, and delight" (Richins, 1997). | ↑ Behavioral intentions towards healthy foods (purchase intentions, willingness to spread positive word-of-mouth, and willingness to recommend the promoted healthy meal) | — | — |
| Jackson et al., 2003 [145] | USA | To validate a measure of psychological motivations to eat based on a four-category model of motivations for alcohol use. Motivations specified by this model are: to cope with negative affect, to be social, to comply with others expectations, and to enhance pleasure. | 520 women, 292 men<br>Mean age: 18.8<br>BMI: N/A | The Pleasure motivation subscale of the Motivation to Eat questionnaire, a questionnaire developed by the authors. | — | ↑ Emotional Eating<br>↑ External Eating | Restrained Eating |
| Le Bel, 2000 [109] | Canada (Québec) | The objectives of study 1 were to model the changes in online ratings (pleasure and desire to consume) associated with pleasures of different intensity, and to study the relationship between pleasure intensity and consumption. | Study 1:<br>22 participants<br>Age range: 25–79<br>Mean age: 35.55 ± 11.36<br>BMI range: 18–30<br>Mean BMI: 23.08 ± 3.55 | Participants provided a hedonic rating on a 100 mm visual analog scale (VAS) anchored with "no pleasure at all 0" and "very intense pleasure 10". | — | — | Consumption of a hedonic food (chocolate)<br>Eating speed |
| Lindeman & Stark, 1999 [74] | Finland | To explore how various food choice motives (including pleasure) cluster together. To explore the relationship between food choice motives and personality, and, in particular, to analyse whether the emerging food choice clusters differ in terms of personality, for example with regard to mental wellbeing. | Study 1:<br>147 women<br>Age range: 17–45<br>Mean age: 25.8 ± 5.8<br>BMI: N/A<br>Study 2:<br>118 girls<br>Age range: 16–18<br>Mean age: 16.44<br>BMI: N/A | Study 1:<br>The 8-item Pleasure subscale of a questionnaire developed by authors and assessing food choice motives.<br>Study 2:<br>The 4-item pleasure subscale (sensory appeal) of The Food Choice Questionnaire (Steptoe et al., 1995). | — | ↑ BMI | Awareness of thinness pressures<br>Internalization of thinness pressures<br>Appearance dissatisfaction<br>Weight dissatisfaction<br>Symptoms of eating disorders<br>Depressive symptoms<br>Self-esteem |
| Lindeman & Stark, 2000 [30] | Finland | To compare four types of dieters who stressed ideological food-choice reasons and pleasure in their food choice in different ways. | 66 women<br>Age range: 14–68<br>Mean age: 33.85 ± 13.62<br>BMI: N/A | The 4-item pleasure subscale (sensory appeal) of The Food Choice Questionnaire (Steptoe et al., 1995). | ↓ Depressive symptoms | — | Symptoms of eating disorders |
| Loh et al., 2013 [158] | Malaysia | To describe the adaptation and validation of the Children's Eating Behavior Questionnaire (CEBQ) as a self-report for adolescents, investigates gender and ethnic differences in eating behavior and examines associations between eating behavior and body mass index (BMI) z-scores among multi-ethnic Malaysian adolescents. | 464 girls, 182 boys<br>Age range: 13<br>BMI: 20.6 ± 5.1<br>BMI z-score: 0.21 ± 0.93<br>27% were overweight or obese | The 4-item "Enjoyment of food" subscale of a self-reported version of the Children's Eating Behavior Questionnaire (CEBQ) developed by the authors. | — | — | BMI |

*(Continued)*

**Table 5.** (Continued)

| Reference | Country | Study aim | Sample | Measurement of eating pleasure | Favorable links | Unfavorable links | Neutral links |
|---|---|---|---|---|---|---|---|
| Marquis & Shatenstein, 2005 [75] | Canada (Québec) | To examine the contribution of immigrant mothers' food motives (including pleasure) to the importance placed on family meals, and cultural differences in mothers' food motives and the importance ascribed to family meals. | 209 mothers Age: N/A BMI: N/A | The 6-item pleasure subscale of a modified version of the Food Choice Questionnaire (FCQ; Steptoe et al., 1995) | ↑ Perceived importance of family meals | — | — |
| Marty et al., 2017 [26] | France | To investigate whether hedonic versus nutrition based implicit and/or explicit attitudes toward food predicts children's healthy versus unhealthy food choices. | 31 girls, 32 boys Age range: 6.3–11.5 Mean age: 8.99 ± 1.51 Mean BMI z-score: 1.65 ± 1.92 BMI z-score range: −1.74–5.74 | Implicit hedonic score: For each triplet of foods, children were asked to "choose the two foods that best go together". The children were able to perform hedonic associations by pairing two food items that are typically consumed together (e.g., steak with fries or chicken with fries) or a nutritional association by pairing the two nutritionally similar food items (e.g., steak with chicken because they both belong to the meat category). An implicit hedonic score based on the percentage of hedonic associations was calculated for each child (range: 0–100%). Explicit hedonic score: Explicit attitudes that were assessed with a categorization task in which children placed food items into one of the following categories: "yummy", "yucky" (i.e., hedonic categories), "makes you strong", or "makes you fat" (i.e., nutritional categories). | ↑ Healthy food choices ↑ Healthy food choices | — | — |
| Mason et al., 2017 [76] | USA | To evaluate the ecological validity (i.e., accurate measurement of a construct as experienced in naturalistic settings) of the self-report Dutch Eating Behavior Questionnaire (DEBQ). | 42 women, 8 men Age: adults BMI: all obese Mean BMI: 40.3 ± 8.5 | During pre-episode recordings, participants rated the item "I will enjoy the taste of this food.". During post-episode recordings, participants rated the item "I enjoyed taste of the food,". | — | ↑ External eating (with predicted pleasure) ↑ Restraint eating (with actual reported enjoyment at post-eating episode rating) | Restraint eating (with predicted pleasure) External eating (with actual reported enjoyment at post-eating episode rating) Emotional eating |
| McClain et al., 2011 [150] | USA | To present the development of the Meanings of Eating Index (MEI) as well as to investigate the meanings of eating behavior and their association with dietary intake in minority youth as a potential avenue for effective dietary intervention. | 60 girls, 40 boys Mean age: 9.52 ± 0.56 BMI: N/A | The Pleasure eating subscale of the Meanings of Eating Index (MEI) questionnaire, a questionnaire developed by the authors. | — | — | Dietary intake |
| Olsen & Tuu, 2017 [33] | Vietnam | To investigate the mediating and dual role of hedonic and healthy eating values in the relationships between consideration of future consequences and convenience food consumption. | 223 girls, 228 boys Age range: 13–19 BMI: 28.8% were overweight or obese | Respondents were asked to express their hedonic eating values on a scale beginning with 'It is important to me that food I eat. . .' and corresponding to five items adapted from previous studies assessing hedonic consumption values (Babin et al., 1994). | — | ↑ convenience food consumption | — |

(*Continued*)

**Table 5.** (Continued)

| Reference | Country | Study aim | Sample | Measurement of eating pleasure | Favorable links | Unfavorable links | Neutral links |
|---|---|---|---|---|---|---|---|
| Phan & Chambers, 2016 [34] | USA | To explore motivations behind eating occasions by looking at specific choices of foods and beverages people consumed at various mealtimes. | 162 women, 36 men<br>Age range: 18 and +<br>BMI: N/A | The 3-item pleasure motivation subscale of the Eating Motivation Survey (TEMS; adapted from Renner et al., 2012). | — | Motivates late-night snaking | — |
| Phan & Chambers, 2016 [13] | USA | To investigate the motivations behind everyday choices of different food groups using a bottom up approach that targeted the specific choices of foods and beverages people consumed at various times of a day and then organized those products into food groups to examine larger trends for that food group. | 162 women, 36 men<br>Age range: 18 and +<br>BMI: N/A | The 3-item pleasure motivation subscale of the Eating Motivation Survey (TEMS; adapted from Renner et al., 2012). | Motivates tea consumption | Motivates sweets consumption<br>Motivates soda consumption<br>Motivates alcoholic beverages consumption | — |
| Remick et al., 2009 [112] | Canada | To examine the relationship between eating pleasure, restrained eating, and well-being. | 96 women, 63 men<br>Mean age: 19.1 ± 1.5<br>BMI: N/A | A 10-item questionnaire assessing consumption attitudes developed by the authors | ↑ Self-clarity (non-restrained individuals)<br>↓ Neuroticism (non-restrained individuals) | ↓ Self-clarity (restrained individuals)<br>↑ Neuroticism (restrained individuals) | Life satisfaction |
| Roininen & Tuorila, 1999 [84] | Finland | To assess the predictive value of the Health and Taste Attitude questionnaires in a simple behavioral test situation, and to compare the results of the choice task with another commonly used measure of behavior, that of reported use frequency of each food item. | 90 women, 54 men<br>Age range: 15–60<br>Mean age: 32<br>BMI: N/A | The 6-item Pleasure subscale of the Health and Taste Attitude questionnaires (Roininen et al., 1999). | — | — | Food choices (healthy vs. unhealthy)<br>Reported use frequency (healthy vs. unhealthy) |
| Roininen et al., 2001 [86] | Finland /UK/ The Netherlands | To compare the Health and Taste Attitude Scales (HTAS) factor structure of Finnish, British and Dutch samples and validate the HTAS cross nationally by examining the relationship between scale scores and reported behavior. | 770 women, 535 men<br>Age range: 18–75<br>BMI: N/A | The 6-item Pleasure subscale of the Health and Taste Attitude questionnaires (Roininen et al., 1999). | — | ↑ intake of full-fat cheese sandwich and full-fat chocolate bar<br>↓ intake of reduced-fat cheese sandwich<br>↑ intake of light soft drink | Intake of reduced-fat chocolate bar<br>Intake of non-fat milk<br>Intake of full-fat milk<br>Intake of regular soft drink<br>Healthy food choice<br>Pleasant choice |
| Smith & Hawks, 2006 [126] | USA | To evaluate the relationship between intuitive eating, diet composition, nutritional quality of diet, and certain meanings associated with food, including food anxieties and pleasure associated with eating. | 136 women, 207 men<br>Age range: 18–26<br>BMI: N/A | The 18-item Pleasure subscale of a questionnaire developed to explore the role of food in life (Rozin et al., 1999) | ↑ Intuitive eating | — | — |
| Somers et al., 2014 [88] | Australia | To examine the associations between socio-demographic, social and hedonic characteristics and food involvement in a sample of Australians aged over 55 years. | 522 women, 519 men<br>Age range: 55–88<br>Mean age: 66 ± 6.99<br>BMI: N/A | The 3 items of the Pleasure motivation subscale from the Health and Taste scales (Roininen et al. 1999). | ↑ Perceived level of importance that individual place on food | — | — |

(Continued)

**Table 5.** (Continued)

| Reference | Country | Study aim | Sample | Measurement of eating pleasure | Favorable links | Unfavorable links | Neutral links |
|---|---|---|---|---|---|---|---|
| Vinai et al., 2016 [152] | Italy | To evaluate if the decision to have seconds is related to the current evaluation of its palatability or to the predicted pleasure of a second helping. | 62 women, 66 men<br>Age range: 19–69<br>Mean age: 42.44 ± 12.34<br>BMI range: 16.77–35.27<br>Mean BMI: 23.21 ± 3.15 | Participants were asked to indicate on a Likert Scale from 0 to 4 the pleasure they experienced from eating their portion of soup and the pleasure they predicted they would experience by having a second helping of the same soup. | — | ↑Decision to have a second helping of an energy-dense, high fat soup (predicted pleasure). | Decision to have a second helping of an energy-dense, high fat soup (experienced pleasure related to the first helping). |
| **Prospective studies (n = 1)** | | | | | | | |
| Thogersen-Ntoumani et al., 2009 [146] | Greece | To examine whether motivation to eat variables predict changes in dieting and weight control behaviors in both gender groups over a 5-month interval. | 148 girls, 99 boys<br>Age range: 14–18<br>Mean age: 14.75 ± 0.76<br>BMI: 15.7% were overweight or obese | The Pleasure motivation subscale of The Motivation to Eat Scale (MES; Jackson et al., 2003) | ↓ Risk of fasting to control or maintain weight (only men) | ↓ Likelihood of eating more fruits and vegetables to lose or maintain weight (only women) | Likelihood of eating more fruits and vegetables to lose or maintain weight (only men)<br>Likelihood of eating less high fat foods to lose or maintain weight<br>Likelihood of eating less sweets to lose or maintain weight<br>Likelihood of fasting to lose or maintain weight (only women)<br>Likelihood of eating very little food to lose or maintain weight<br>Likelihood of using a food substitute to lose or maintain weight<br>Likelihood of skipping meals to lose or maintain weight<br>Likelihood of vomiting to lose or maintain weight |
| **Short-term, single exposure intervention studies (n = 8)** | | | | | | | |
| Arch et al., 2016 [140] | USA | To examine in a laboratory context whether brief mindfulness offers both psychological and physical benefits by enhancing the positive experience of eating while simultaneously decreasing caloric consumption. | Study 1:<br>48 women, 33 men<br>Age range: 18–26<br>Mean age: 19.49 ± 1.80<br>BMI: N/A<br>Study 2:<br>104 women, 32 men<br>Age range: 18–35<br>Mean age: 20.1 ± 2.4<br>BMI: N/A<br>Study 3:<br>43 women, 59 men<br>Age range: 18–37<br>Mean age: 20.78 ± 3.87<br>BMI: N/A | Study 1 and 2: Participants were asked: "Using any number between 1 (hated it) and 10 (loved it), please indicate how much you enjoyed tasting the chocolate."<br>Study 3: How much do you enjoy eating this food? (with 5 anchors, including 0 = not at all to 7 = extremely) | The third study showed that brief mindfulness instructions led to lower consumption of unhealthy food calories relative to distracted or no-instruction control conditions, an effect mediated by greater eating enjoyment. | — | Consumption of healthy food calories |

Relative to distraction control instructions, the first two studies demonstrated that brief mindfulness instructions increased the enjoyment of a commonly pleasurable food (chocolate; Study 1), and a food with generally more mixed associations (raisins; Study 2).

*(Continued)*

Table 5. (Continued)

| Reference | Country | Study aim | Sample | Measurement of eating pleasure | Favorable links | Unfavorable links | Neutral links |
|---|---|---|---|---|---|---|---|
| Cornil & Chandon, 2016 [37] | France/USA | To investigate how multisensory imagery can make people happier with smaller food portions. The intervention asks people to vividly imagine the multisensory pleasure (taste, smell, and texture) of three hedonic foods before choosing the size of another hedonic food. This intervention was compared to several control conditions (e.g, non-food sensory imagery condition). | Study 1: 22 girls, 20 boys Age: 5 BMI: 0% were obese Study 2: 124 women, 76 men Mean age: 34 BMI: N/A Study 3: 60 women, 40 men Mean age: 34 BMI: N/A Study 4: 367 women Mean age = 22 BMI: N/A Study 5: 114 women, 76 men Mean age: 37 BMI: N/A | Not measured | Compared with control conditions, focusing on sensory pleasure makes hungry eaters and nondieters choose and actually prefer smaller portions of hedonic foods. | Compared with control conditions, focusing on sensory pleasure makes sated eaters and dieters choose larger portions of hedonic foods. | — |
| Hege et al., 2018 [159] | Germany | To explore behavioral responses and neural processes during pre meal planning. In particular, to investigate whether different mindsets (healthiness mindset, pleasure mindset, fullness mindset) were associated with altered activity in certain brain areas during the pre-meal selection of portion sizes. | 9 women, 9 men Age range: 18–31 Mean age: 24.6 BMI range: 19.5–24.0 Mean BMI: 21.8 | Not measured | Compared with a free choice condition, participants reduced their portion sizes when considering eating for health or pleasure while increasing their portion sizes when considering eating to be full until the next meal. | — | — |
| Hong et al., 2014 [141] | USA | The goal of the present research was to more clearly test the hypothesis that mindful eating creates higher enjoyment of eaten food and to explore whether mindful eating is associated with more willingness to sample novel or typically avoided foods. | 253 women; 158 men Age range: 17–40 Mean age: 18.92 ± 1.01 BMI: N/A | Participants reported how much they enjoyed the food using a 7-point scale (1 = extremely negative and found the food extremely unenjoyable, 4 = neutral, 7 = extremely positive and found the food extremely enjoyable). Samplers in the mindful raisin-eating task condition indicated higher levels of enjoyment of the sampled food than did samplers in either the nonmindful raisin-eating control condition or the no-task baseline condition. | — | — | Mindful eating had no effect on the likelihood of sampling food. The mindfulness manipulation produced no differences in participants' mood. |

(Continued)

**Table 5.** (Continued)

*(Continued)*

| Reference | Country | Study aim | Sample | Measurement of eating pleasure | Favorable links | Unfavorable links | Neutral links |
|---|---|---|---|---|---|---|---|
| Huang & Wu, 2016 [125] | USA | Previous research has shown a "healthy = less tasty" intuition. This intuition leads consumers to perceive a food with a healthy name (e.g., salad) as less tasty than the same food with an unhealthy name (e.g., pasta). The present study demonstrated that this effect of product name diminished or even reversed when consumers were high (vs. low) in food pleasure orientation. | 165 women, 131 men Mean age: 36.69 ± 11.84 BMI: 23.0% were obese | Food pleasure orientation was measured using 6 items from the questionnaire of Rozin et al. (1999) that explores the role of food in life. | In contrast to those with low food pleasure-orientation, high food pleasure-oriented participants rated food with a healthy name as tastier than food with an unhealthy name. Compared to low food pleasure-oriented individuals, high food pleasure-oriented individuals did not show the tendency to choose a high-calorie dessert after they imagined having eaten something healthy. | — | — |
| Petit et al., 2016 [7] | France | To assess whether drawing attention to the tastiness of healthy food modulates activity in brain areas that are used to exercise self-control and whether it increases healthy food choices in individuals with a high BMI. Participants were shown a picture of one of the 64 selected food items and were given up to 3 s to indicate whether they would be willing to eat that item at the end of the experiment or not. Participants evaluated each food item in three attention conditions (healthy diet, tasty diet, and no diet). In the healthy diet condition, the participants were instructed to consider the benefits of eating healthy food. In the tasty diet condition, they were instructed to consider the pleasure of eating healthily, and in the no diet condition (control condition), they were asked to consider any features of the food that came to their mind. | 10 women, 13 men Mean age: 25.91 ± 3.85 Mean BMI: 23.47 ± 2.8 | Not measured | The percentage of the time that participants responded "yes" or "strong yes" to eating healthy food was significantly higher in the healthy diet condition and the tasty diet condition than in the no diet condition. The percentage of healthy food choices was not significantly different between the healthy diet and tasty diet conditions. | — | — |
| Robinson et al., 2011 [147] | UK | To examine the relationship between remembered enjoyment of vegetable eating, predictions about liking for vegetables, and vegetable consumption. | Study 1: 38 women, 16 men Mean age: 22.0 ± 2.9 BMI: N/A Study 2: 66 women, 29 men Mean age: 22.0 ± 3.7 Mean BMI: 24.2 ± 3.5 Study 3: 50 women, 13 men Mean age: 19.3 ± 1.3 Mean BMI: 23.2 ± 3.5 | Enjoyment of the recall memories were rated using a 10-cm line scale, anchors: "strongly disliked," on the left and "strongly liked," on the right. | When people recalled positive and enjoyable memories of past vegetable eating it resulted in higher liking for vegetables and choice of a larger portion size of vegetables compared with recalling a personal non-food memory, a non-vegetable food memory, or visualization of someone else enjoying eating vegetables (increase of approximately 70% in vegetable portion size compared to controls). | — | — |

**Table 5.** (Continued)

| Reference | Country | Study aim | Sample | Measurement of eating pleasure | Favorable links | Unfavorable links | Neutral links |
|---|---|---|---|---|---|---|---|
| Robinson et al., 2012 [148] | UK | To examine if a simple intervention (simple act of rehearsal) could be used to increase remembered enjoyment of a food and whether this would result in individuals choosing to eat more of the food at a later date. | Study 1: 48 women, 10 men Mean age: $20.2 \pm 3.2$ Mean BMI: $23.3 \pm 4.4$ Study 2: 32 women, 5 men Mean age: $20.1 \pm 2.8$ Mean BMI: $22.6 \pm 3.9$ | Study 1: Participants answered three questions assessing remembered enjoyment using separate 10 cm line scales. (1) 'Compared to an average lunch, yesterday's lunch was'; anchors (from left to right)–'not at all enjoyable' and 'extremely enjoyable'. (2) 'I would enjoy eating the meal again'; anchors–'not at all likely' and 'extremely likely'. (3) 'I would recommend the meal to a friend'; anchors–'not at all likely' and 'extremely likely'. Study 2: Participants were asked to think back to the foods eaten and rate how enjoyable they were, using a 10 cm visual analogue scales for each food; anchors (from left to right)–'not at all enjoyable' and 'extremely enjoyable.'. Results showed that, compared with control conditions, remembered enjoyment can be increased via a simple act of rehearsal. | Study 2 showed that, compared with control conditions, remembered enjoyment resulted in a later 90% increase in the amount of low-energy meal (Mediterranean vegetable quiche) chosen and eaten. | — | — |
| **Longer-term, prolonged exposure intervention studies (n = 5)** | | | | | | | |
| Adam et al., 2015 [160] | International | To assess the effectiveness of open, online nutrition and cooking instruction in improving the eating behaviors of course participants. | 6382 female, 979 male, 24 others Age range 13 and + BMI: 32.8% perceived themselves to be overweight or obese | Levels of enjoyment of the previous night's dinner rated on five-point Likert scales and categorized as follows: not enjoyable or somewhat enjoyable (coded 0), moderately enjoyable (coded 1), and very or extremely enjoyable (coded 2). There was a significant increase in the percentage of participants who found yesterday's dinner to be very or extremely enjoyable in response to the intervention. | Almost all pre/post survey comparisons showed significant changes in the desired direction for eating behaviors and meal composition, mirroring the messaging of the course (e.g., cooking at home with mostly fresh foods, increase in consumption of fresh vegetables and fresh fruits). Mean overall perceived ease of making healthy food choices and preparing home cooked meals also increased significantly over time. | — | Little change was seen from pre to post-course survey with regard to the consumption of foods that had neither been promoted nor discouraged in course videos (e.g., dairy). |

*(Continued)*

**Table 5.** (Continued)

| Reference | Country | Study aim | Sample | Measurement of eating pleasure | Favorable links | Unfavorable links | Neutral links |
|---|---|---|---|---|---|---|---|
| Ensaff et al., 2016 [136] | UK | To examine Jamie Oliver's Kitchen Garden Project (JOKGP), involving timetabled food sessions in a purpose-built kitchen classroom. | Pupil: baseline 325, follow-up 338 Parent: baseline 79, follow-up 73 Pupils: Age range: 7–11 BMI: N/A Parents: Age range: N/A BMI: N/A | Items related to food enjoyment were categorical and presented on a 4-point Likert scale ranging from "never" to "always" providing a range of 1–4. Cooking enjoyment was measure using the item "Do you like cooking"? In the intervention school, pupils reported an increase in cooking enjoyment. | In the intervention school, pupils reported an increase in helping with cooking at home, taste description score, and a decrease in food neophobia and fussiness. At follow-up, higher scores were reported for the pupils in the intervention school compared with pupils from the control school for cooking experience and taste description. They were also three times as likely to report chopping vegetables or fruit and eight times as likely to report using a zester. Data collected from parents revealed higher scores for these two questions: 'Does your child help with cooking at home?' and 'Does your child ever ask you to make food that he/she has tried at school?' In addition, the response to the item 'Has your child introduced you to new (unfamiliar to you) foods or dishes?' was significantly different, with those parents from the intervention school more likely to respond "Yes". | — | Despite increases observed in the intervention school, no difference between schools (intervention vs. control) was observed for Total Kitchen Help, the item "Do you like cooking?", the item "Do you help with cooking at home?", and the Food Score and Food Neophobia & Fussiness Score at follow-up. |
| Gravel et al., 2014 [31] | Canada (Quebec) | To investigate and determine whether sensory-based intervention influenced eating related attitudes and behaviors among restrained women, as well as reliance on physical signals for hunger and satiety. A registered dietitian conducted a sensory-based intervention during six weekly 90-min workshops. Specific themes were addressed in each of the six workshops, the sixth exploring the pleasures associated with eating (such as biological, social, emotional, and cultural). | 50 restrained women Age range: 25–60 Mean age: 47.5 ± 10.0 Mean BMI: 27.7 ± 5.9 | Not measured | The sensory-based intervention decreased disinhibition, situational susceptibility to disinhibition, and increased unconditional permission to eat (Intuitive eating subscale) compared with a waiting list control group. | — | Restraint eating Cognitive dietary restraint (flexible and rigid control) Habitual susceptibility to disinhibition Emotional susceptibility to disinhibition Susceptibility to hunger (internal and external locus) Mindful Eating Questionnaire subscales (except disinhibition) Intuitive Eating Subscales (except Unconditional Permission to Eat) BMI |
| Gravel et al., 2014 [161] | Canada (Quebec) | To investigate and determine whether sensory-based intervention influenced the number and type of terms (descriptive and hedonic) used by restrained women to describe a certain food, and whether changes in the number of descriptive terms were associated with changes in intuitive eating. A registered dietitian conducted a sensory-based intervention during six weekly 90-min workshops. Specific themes were addressed in each of the six workshops, the sixth exploring the pleasures associated with eating (such as biological, social, emotional, and cultural) | 50 restrained women Age range: 25–60 Mean age: 47.5 ± 10.0 Mean BMI: 27.7 ± 5.9 | Not measured | The sensory-based intervention increased the use of descriptive terms associated with all senses (except for sight-related terms) and the ratio of descriptive vs. hedonic terms compared with a waiting list control group. | — | Hedonic terms associated with all senses |

*(Continued)*

**Table 5.** (Continued)

| Reference | Country | Study aim | Sample | Measurement of eating pleasure | Favorable links | Unfavorable links | Neutral links |
|---|---|---|---|---|---|---|---|
| Sasson et al., 2007 [32] | USA | Little research has assessed the efficacy of nutrition education curricula that emphasize the importance of enjoying the cooking and eating experience, taking time to eat, and valuing high-quality, unprocessed food. Furthermore, little is known about how long-term outcomes of such an approach would compare with other more popular yet restrictive dietary approaches. The 3-week NYU Study Abroad Program in Italy consists of participants examining food and nutrition from historical, cultural, and culinary perspectives. | 15 women, 1 man Mean age: 27 BMI: N/A | Not measured | The program positively influenced behaviors related to food shopping, preparation, eating, and dieting even 6 months later. More precisely, 63% of participants reported increases in purchases of locally grown food, 81% and 50% reported increases in purchases of seasonally and organically grown food, 50% reported more frequent monthly purchases of food from farmers' markets, 50% reported more frequent monthly purchases of olive oil, 75% reported increases in purchases of wine and wine consumption with meals, 81% reported increasing weekly wine consumption with meals, 50% reported an increase in the frequency of consuming "more satisfying" meals, 31% reported decreasing portion sizes whereas 6% reported an increase, 37% reported increasing "the amount of time I spend eating meals", more than 50% reported changes in eating behaviors. The number of meals prepared at home, the use of fresh ingredients, trying new recipes, interest in improving cooking skills, the variety of foods used to prepare food, and the use of olive oil in cooking were all increased by the majority of participants. In addition, 31% reported that they have decreased "the amount that I worry about calories" whereas 6% reported an increase in this behavior, and more than 80% reported an increase for "the amount that I make time for cooking, eating and enjoying food". | — | Self-reported weight |

the role of food in life (Rozin et al., 1999) [125,126], (18) the food enjoyment subscale of a questionnaire assessing children's relationship with food (Ensaff et al., 2016) [136] and (19) a questionnaire assessing remembered enjoyment (Robinson et al. 2011) [148]. Nine studies [21,26,28,74,76,136,140,148,152] included multiple measures of eating pleasure while six intervention studies [7,31,32,37,159,161] that used eating pleasure in their strategies did not measure the impact of their interventions on eating pleasure.

**Dietary behavior and health outcomes.** Dietary behavior and health outcomes for each study are presented in Table 5. Of the 45 studies identified, 31 (68.9%) included only dietary outcomes [7,13,25–29,33,34,37,75,76,84,86,88,109,111,125,126,136,140,145–148,150–152,159,160,161], 10 (22.2%) only health outcomes [30,35,36,74,89,112,137,156–158] and 4 (8.9%) included both dietary and health outcomes [21,31,32,141].

The dietary outcomes studied were very diverse, the most studied being diet quality (i.e., food/dietary intakes, balanced diet, adherence to nutritional guidelines) (n = 13) [13,27–29,32,33,84,86,109,140,148,150,160], food choices (n = 8) [7,26,84,86,125,147,148,160], portion size (n = 4) [21,32,37,159], and restrained eating (n = 4) [21,31,76,145]. Other dietary outcomes were assessed in two studies or less and comprized emotional eating (n = 2) [76,145], external eating (n = 2) [76,145], intuitive eating (n = 2) [31,126], perceived importance of family meals (n = 2) [75,111], food neophobia/fussiness (n = 2) [136,141], taste description ability (n = 2) [136,161], likelihood to try new food / recipes (n = 2) [32,136], cooking skills (n = 2) [136,160], the amount time spent for cooking/involvement in cooking activities (n = 2) [32,136], the amount of time spent eating meals (n = 2) [32,109], subjective diet-related quality of life (n = 1) [29], perceived level of importance that individuals place on food (n = 1) [88], skipping breakfast habit (n = 1) [29], energy intake (n = 1) [27], behavioral intentions toward healthy foods (n = 1) [151], late-night snacking (n = 1) [34], family meals frequency (n = 1) [111], eating habits to control weight (n = 1) [146], perceived ease of making healthy food choices (n = 1) [160], disinhibition (n = 1) [31], susceptibility to hunger (n = 1) [31], mindful eating (n = 1) [31], worries about calories (n = 1) [32], nutritional status (n = 1) [25], alcohol intake (n = 1) [32], frequency of consuming satisfying meals (n = 1) [32], the number of meals prepared at home (n = 1) [32], the use of fresh ingredients (n = 1) [32], interest in improving cooking skills (n = 1) [32], the variety of foods used to prepare food (n = 1) [32], and the purchases of locally grown food, seasonally and organically grown food and healthy food (n = 1) [32].

The health outcomes were also diverse, the most studied being body weight/BMI (n = 8) [21,31,32,35,36,74,157,158] and depressive symptoms (n = 3) [30,74,156]. Other health outcomes assessed in two studies or less were happiness (n = 2) [89,137], satisfaction with life (n = 2) [112,156], symptoms of eating disorders (n = 2) [30,74], anxiety symptoms (n = 1) [156], well-being (n = 1) [21], health worries (n = 1) [21], awareness of thinness pressures (n = 1) [74], internalization of thinness pressures (n = 1) [74], appearance dissatisfaction (n = 1) [74], weight dissatisfaction (n = 1) [74], self-esteem (n = 1) [74], self-clarity (n = 1) [112], neuroticism (n = 1) [112] and mood (n = 1) [141].

**Links between eating pleasure and dietary/health outcomes.** Summary of associations between eating pleasure and dietary and health outcomes are shown in **Fig 2**.

Among the 35 studies that included dietary outcomes, 20 (57.1%) were cross-sectional studies, eight (22.9%) were short-term, single exposure intervention studies, five (14.3%) were longer-term, prolonged exposure intervention studies, one (2.9%) was a mixed-design study and one (2.9%) was a prospective study. Of these, 20 (57.1%) observed favorable associations between eating pleasure and dietary variables [7,25,26,28,29,32,75,88,111,125,126,136,140,147,148,151,159–161] while six (17.1%) reported unfavorable associations [33,34,76,86,145,152], five (14.3%) mixed associations (both favorable

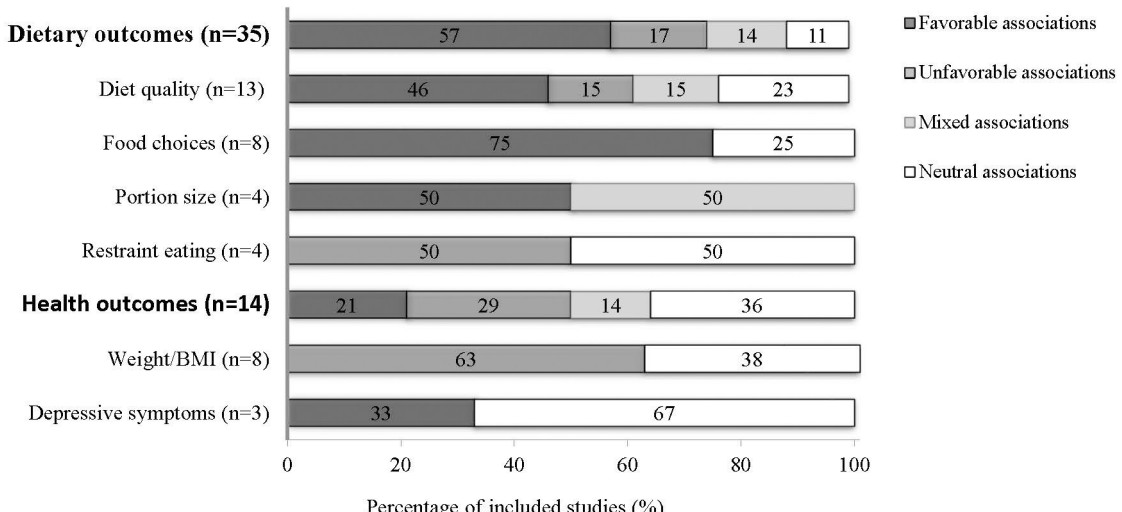

**Fig 2. Links relating eating pleasure with dietary and health outcomes in the scientific literature.**

and unfavorable associations) [13,21,27,37,146] and four (11.4%) neutral association [84,109,141,150]. When only intervention studies were considered, we observed that most of the papers (11/13; 84.6%) reported positive dietary outcomes [7,32,125,136,140,147,148,159–161], while only one observed mixed effects [37] and one neutral effect [141].

When examining diet quality specifically among the different dietary outcomes (n = 13), six studies (46.2%) observed favorable associations between eating pleasure and diet quality [28,29,31,32,140,148] while two (15.4%) reported unfavorable associations [33,86], two (15.4%) mixed associations [13,27] and three (23.1%) reported neutral association [84,109,150]. For food choices (n = 8), six studies (75.0%) observed favorable associations [7,26,125,147,148,160] while two (25.0%) reported neutral association [84,86]. For portion size (n = 4), two studies (50.0%) observed favorable associations [32,159] while two (50.0%) reported mixed associations [21,37]. Finally, for restrained eating, two studies (50%) reported unfavorable associations [21,76] while two (50.0%) observed neutral association [31,145]. Other dietary outcomes were assessed in two studies or less, therefore separated analysis was not performed for these variables.

Among the 14 studies that documented health outcomes, nine (64.3%) were cross-sectional studies, two (14.3%) were longer-term, prolonged exposure intervention studies, two (14.3%) were qualitative studies and one (7.1%) was a short-term, single exposure intervention study. Of these, three (21.4%) observed favorable associations between eating pleasure and health variables [30,89,137] while four (28.6%) reported unfavorable associations [35,36,74,157], two (14.3%) mixed associations [21,112] and five (35.7%) neutral association [31,32,141,156,158] (Fig 2). Only three intervention studies measured health outcomes and they reported no impact of the eating pleasure-related interventions on these variables.

Most studies linked eating pleasure to a higher weight/BMI (n = 5/8; 62.5%) [21,35,36,74,157], while three studies reported neutral association (37.5%) [31,32,158]. In addition, most studies reported neutral association between eating pleasure and depressive symptoms (n = 2/3; 66.7%) [74,156] while one study reported a favorable association (33.3%) [30]. Other health outcomes were assessed in two studies or less, therefore separate analysis was not performed for these variables.

**Links between dimensions of eating pleasure and dietary/health outcomes.** For each article, up to five dimensions of eating pleasure were identified by coders. In order of

**Table 6. Links between eating pleasure and dietary/health outcomes according to the dimensions used to conceptualize eating pleasure.**

| Dimensions of eating pleasure | Dietary outcomes | | | | Health outcomes | | | |
|---|---|---|---|---|---|---|---|---|
| | Favorable | Unfavorable | Mixed | Neutral | Favorable | Unfavorable | Mixed | Neutral |
| Sensory experiences | [7,21,25,26,75,88] | [33,76,86] | [37] | [84,109] | [21,30,89] | [74] | | |
| Social experiences | [29,75,111,125,126] | | [27] | | | [36] | [112] | |
| Food characteristics | [75] | | [27] | | | [36] | | |
| Food preparation process | [28,136] | | | | | | | |
| Novelty | | [33] | [27] | | [89,137] | [36] | | |
| Variety | | | [27] | | [137] | [36] | | |
| Mindful eating | [140] | | | [141] | | | | [141] |
| Visceral eating | | [21,34,145] | [13,146] | | | [21] | | |
| Place | [126] | | | | | | | |
| Memories | [125,126,147,148] | | | | [89] | | [112] | |
| Atmosphere | | | | | | | | |
| Psychological / physical state during food intake | [151] | [33] | | [150] | [137] | | | |
| Food anticipation | [125,126,151] | [152] | | | | | [112] | |
| Special occasions | | | | | | | | |
| Having the choice | | | | | | | | |
| Food intake structure | [28] | | | | | | | |
| Taking time | | | | | [137] | | | |
| Health considerations | | | | | | | | |
| Food preferences | | | [27] | | | [36] | | |
| Psychological / physical state after food intake | [75] | | | | | | | |
| Eating habits | | | | | | | | |
| Ideological considerations | | | | | | | | |

importance, dimensions studied in relation to dietary outcomes were sensory experiences (n = 12) [7,21,25,26,33,37,75,76,84,86,88,109], social experiences (n = 6) [27,29,75,111,125,126], visceral eating (n = 5) [13,21,34,145,146], memories associated with eating (n = 4) [125,126,147,148], food anticipation (n = 4) [125,126,151,152], psychological and physical state during food intake (n = 3) [33,150,151], food characteristics besides sensory attributes (n = 2) [27,75], food preparation process (n = 2) [28,136], novelty (n = 2) [27,33], mindful eating (n = 2) [140,141], variety (n = 1) [27], the place (n = 1) [126], food intake structure (n = 1) [28], food preferences (n = 1) [27], and psychological and physical state after food intake (n = 1) [75] (**Table 6**). No study relating eating pleasure to dietary outcomes conceptualized eating pleasure in terms of atmosphere, eating on special occasions, having the choice, taking time, health considerations, eating habits or ideological considerations.

In order to evaluate associations between dimensions of eating pleasure and dietary outcomes, we focused on dimensions identified in at least two different documents (Table 6). We found that when eating pleasure was conceptualized using social experiences, 5 out of 6 studies (83.3%) [29,75,111,125,126] showed that eating pleasure was associated with positive dietary outcomes while the percentage of studies showing positive dietary outcomes reached 100% when eating pleasure was conceptualized using food preparation process [28,136] or memories associated with eating [125,126,147,148]. When eating pleasure was conceptualized as food anticipation, 3 out of 4 studies (75.0%) [125,126,151] showed that eating pleasure was associated with positive dietary outcomes. However, when pleasure was conceptualized using the visceral eating dimension, it was mostly associated with negative outcomes (3/5; 60.0%) [21,34,145].

Dimensions of eating pleasure studied in relation to health outcomes were fewer in number than when studying dietary outcomes (Table 6). In order of importance, dimensions were sensory experiences (n = 4) [21,30,74,89], novelty (n = 3) [36,89,137], social experiences (n = 2) [36,112], variety (n = 2) [36,137], memories associated with eating (n = 2) [89,112], food characteristics (n = 1) [36], mindful eating (n = 1) [141], visceral eating (n = 1) [21], psychological and physical state during food intake (n = 1) [137], food anticipation (n = 1) [112], taking time (n = 1) [137], and food preferences (n = 1) [36]. Among those studies, none conceptualized eating pleasure in terms of food preparation process, place, atmosphere, eating on special occasions, having the choice, food intake structure, health considerations, psychological and physical state after food intake, eating habits or ideological considerations.

In order to evaluate associations between dimensions of eating pleasure and health outcomes, we focused on dimensions identified in at least two different documents. When eating pleasure was conceptualized as sensory experiences, 3 out of 4 studies (75%) [21,30,89] showed that eating pleasure was associated with positive health outcomes. When eating pleasure was conceptualized as novelty, 2 out of 3 studies (66.7%) [89,137] showed that eating pleasure was associated with positive health outcomes. No dimension identified in at least two documents was predominantly associated with negative outcomes.

## Q. 3 –Most promising intervention strategies using eating pleasure to promote healthy dietary behaviors

**Description of included documents.** Among the 13 intervention studies identified in this review (Table 5), one was excluded from this analysis since it did not permit identification of intervention strategies using eating pleasure to promote healthy dietary behaviors [125]. In addition, two studies reported results from the same long-term intervention [31,161]. Therefore, 12 intervention papers were included in this analysis [7,32,37,136,140,141,147,148,159–161], reporting results from 11 independent interventions. Seven (63.6%) were short-term, single exposure interventions while four (36.4%) were longer-term, prolonged exposure interventions. Studies were all published after 2007. Interventions were conducted in the United States of America (n = 3; 27.3%), United Kingdom (n = 3; 27.3%), France (n = 1; 9.1%), Canada (n = 1; 9.1%), Germany (n = 1; 9.1%) or in multiple countries (n = 2; 18.2%). In sum, these intervention studies included 10,243 participants, of whom 7,835 (76.5%) were women, 1,569 (15.3%) were men and 839 (8.2%) for whom the sex was not specified. Sample size ranged from 16 to 7,385 participants. Nine were conducted in young and middle-aged adults only (81.8%), one in children only (9.1%) and one in both children and young and middle-aged adults (9.1%).

**Intervention strategies using eating pleasure to promote healthy dietary behaviors.** First, two studies investigated whether inducing a pleasure mindset influences portion sizes and food choices. Hege et al. [159] investigated whether different mindsets (healthiness mindset, pleasure mindset, fullness mindset) were associated with altered activity in certain brain areas during the pre-meal selection of portion sizes. Results showed that, compared with a control condition, participants reduced their portion sizes when considering eating for health or pleasure while increasing their portion sizes when considering eating to be full until the next meal. Petit et al. [7] assessed whether drawing attention to the pleasure of eating healthily increased healthy food choices in individuals with a high BMI. Participants evaluated pictures of 64 selected food items in three different "attention conditions" in which participants were either instructed to consider the benefits of eating healthy food (health condition), to consider the pleasure of eating healthily (pleasure condition), or to consider any features of the food that came to their mind (no diet condition). Results showed that participants were more likely

to mention that they would eat the pictured healthy foods in the pleasure diet condition than in the no diet condition. Taken together, these results suggest that, compared with a control condition, a pleasure mindset may have favorable effects on portion sizes and food choices.

Four intervention studies investigated whether focusing on sensory sensations influences dietary intakes, eating behaviors and food choices. Gravel et al. [31,161] investigated whether a sensory-based 6-week intervention influenced eating related attitudes and behaviors among restrained women (dieters). Specific themes addressed were the relationship with food, hunger and satiety cues, sensory sensations, and the pleasures associated with eating. Compared with a waiting list control group, women in the sensory-based intervention decreased disinhibition (i.e., the tendency to overeat in response to cognitive or emotional cues), while increasing the intuitive eating subscale related to the unconditional permission to eat and the use of terms associated with the senses and with pleasure to describe food. No effects on restrained eating, susceptibility to hunger, and BMI were observed. Moreover, Cornil and Chandon [37] investigated how multisensory imagery can make people happier with smaller food portions. During the intervention, participants were asked to vividly imagine the multisensory pleasure (taste, smell, and texture) of three hedonic foods before choosing the size of another hedonic food. This short intervention led to mixed results, showing that, compared with several control conditions (e.g., non-food sensory imagery condition), focusing on sensory pleasure made hungry eaters and nondieters to choose and prefer smaller portions of hedonic foods while sated eaters and dieters chose larger portions. Two interventions using mindful eating strategies and focusing on the sensory properties of food were also performed. Arch et al. [140] showed that directing participants' attention as fully as possible towards the pleasure associated with the sensory experience of eating increased the enjoyment of a commonly pleasurable food (chocolate), and a food with generally more mixed associations (raisins), and led to lower calorie consumption of unhealthy food, an effect mediated by greater eating enjoyment as compared with a control condition. Finally, in a study by Hong et al. [141] participants were randomly assigned to a mindful raisin-eating task condition (i.e., subjects instructed to be nonjudgmental and fully aware, with all senses, of different aspects of the raisin), a non-mindful raisin-eating task (control) condition, or a no-task baseline condition. All participants were thereafter offered different types of food that they could sample. Subjects in the mindful raisin-eating task condition indicated higher levels of enjoyment of the sampled food than did subjects in the two other conditions. However, mindful eating had no effect on the likelihood of trying novel or typically avoided foods. In sum, results suggest that sensory-based interventions can have positive impact on eating behaviors and portion sizes. Mindful eating strategies focusing on the sensory properties of food also seem to increase eating enjoyment and to lower energy intake from unhealthy food while having no impact on the likelihood to sample a novel or typically avoided food.

Two interventions included aspects related to food preparation and/or sharing. Adam et al. [160] assessed the effectiveness of open, online nutrition and cooking instruction in improving eating behaviors of participants. Results showed that, in addition to increase the percentage of participants who found yesterday's dinner to be very or extremely enjoyable, the intervention resulted in significant changes in eating behaviors and meal composition, mirroring the messaging of the course. Mean overall perceived ease of making healthy food choices and preparing home cooked meals also increased significantly over time. Ensaff et al. [136] evaluated Jamie Oliver's Kitchen Garden Project (JOKGP). This project aims at developing food and cooking skills in children, increase their willingness to try new foods, and gain a better understanding of where food comes from. In the intervention school, children reported an increase in cooking enjoyment. They also reported an increase in helping with cooking at home and taste description score, and a decrease in food neophobia and fussiness. At follow-up, higher

scores were reported for the children in the intervention school compared with those from the control school. Taken together, these results suggest that interventions including cooking lessons and sharing increase the enjoyment of eating and cooking, in addition to improving eating behaviors, diet quality, taste description, and involvement in cooking.

Two studies performed by Robinson et al. [147,148] included aspects related to memories. First, they examined the relationship between remembered enjoyment of vegetable eating, predictions about liking vegetables, and vegetable consumption [147]. Results showed that the recall of positive and enjoyable memories about past vegetable consumption resulted in higher liking for vegetables and the choice of a larger portion size of vegetables when compared with recall of a personal non-food memory, a non-vegetable food memory, or when visualizing someone else enjoying eating vegetables. They then examined if a simple instruction to rehearse what participants found enjoyable about a food immediately after eating it could be used to increase remembered enjoyment of a food (a Mediterranean vegetable quiche) and whether this would result in individuals choosing to eat more of this food at a later date [148]. Compared with controls (i.e., neutral meal rehearsal of the ingredients in the vegetable quiche and how long it took to eat it), participants in the intervention group showed a 90% increase in the amount of low-energy Mediterranean vegetable quiche chosen and eaten. In sum, these results suggest that recalling enjoyable eating memories related to healthy foods may increase the portion size of these foods later.

Finally, Sasson et al. [32] assessed the effects of a 3-week study program conducted in Italy that emphasizes the importance of enjoying the cooking and eating experience, taking time to eat, and valuing high-quality, unprocessed food. The program positively influenced behaviors related to food shopping, preparation, eating, and dieting six months later. More precisely, a majority of participants (i.e., 50% or more) reported increases in purchases of olive oil, locally grown food, seasonally and organically grown food, and food from farmers' markets. The consumption of more satisfying meals, the number of meals prepared at home, the use of fresh ingredients, the interest in improving cooking skills, the variety of foods used to prepare food, the use of olive oil in cooking and trying new recipes were also all increased by the majority of participants. Decreases in portion sizes and in worrying about calories were also observed. This program had no impact on self-reported weight. These results suggest that a more comprehensive intervention using eating pleasure as the core strategy may have many beneficial effects on dietary behaviors.

A summary of findings is presented in **Table 7**. This table groups together the dimensions based on their likely favorable or unfavorable impact and presents the dimensions in an order based on the level of evidence. Also, where applicable, intervention strategies that have shown promising effects for each dimension are also presented.

## Discussion

Many authors now claim that eating pleasure is a promising approach to fostering healthy dietary behaviors [19,31,69,142,162] and some dietary guidelines, such as Canada's Food Guide [22], now include the enjoyment of food among their healthy eating recommendations. In spite of this, until now, no comprehensive review assessing the links between eating pleasure and dietary behaviors was available, making difficult to determine whether and how the pleasure of eating can be a lever for healthy eating and health. Our review contributes to filling some important gaps. First, it provides a more comprehensive overview of the key dimensions of eating pleasure, which may serve as a foundation for researchers interested in studying this concept. This literature review also mapped the evidence to date on the links relating eating pleasure with healthy eating and health, a key step for determining whether the pleasure of

**Table 7. Summary of findings: Dimensions of eating pleasure with favorable or unfavorable links with dietary/health outcomes, level of evidence and promising intervention strategies.**

| Key dimensions | Level of evidence [a] | Promising intervention strategies |
|---|---|---|
| **Dietary outcomes** | | |
| **Favorable** | | |
| **Sensory experiences** | Level II | Food-tasting activities; multisensory imagery |
| **Mindful eating** | Level II | Mindful eating strategies focusing on the sensory experience of eating |
| **Memories** | Level II | Rehearsal of enjoyable eating memories related to healthy foods |
| **Social expériences** | Level III | Sharing activities |
| **Food preparation process** | Level III | Nutrition and cooking lessons; cooking experiences |
| **Food anticipation** | Level V | — |
| **Unfavorable** | | |
| **Visceral eating** | Level V | — |
| **Health outcomes** | | |
| **Favorable** | | |
| **Sensory experiences** | Level V | — |
| **Novelty** | Level V | — |

[a] Level I: Evidence from a systematic review of all relevant randomized controlled trials (RCT), or evidence-based clinical practice guidelines based on systematic reviews of RCT; Level II: Evidence obtained from at least one well-designed RCT; Level III: Evidence obtained from well-designed controlled trials without randomization, quasi-experimental; Level IV: Evidence from well-designed case-control and cohort studies; Level V: Evidence from a systematic review of descriptive and qualitative studies; Level VI: Evidence from a single descriptive or qualitative study; Level VII: Evidence from the opinion of authorities and/or reports of expert committees.

eating can be a potential ally in the promotion of healthy eating. This review also identified strategies using eating pleasure that seem promising to include in interventions aimed at promoting healthy eating. However, some major gaps remain in our understanding of how to use eating pleasure in the promotion of healthy eating habits, which we discuss below.

In total, 110 studies were examined for their way to conceptualize eating pleasure, including both scientific and grey literature. The review of the literature suggests that eating pleasure is a multidimensional concept. Indeed, in our qualitative assessment, two independent coders identified 22 different key dimensions used to define eating pleasure. The dimensions most frequently cited were sensory experiences, social experiences, food characteristics besides sensory attributes, food preparation process, novelty, variety, mindful eating, visceral eating, place where food is eaten, and memories associated with eating. This way to conceptualize eating pleasure is much broader than those found in previous reviews, which usually address only one or two dimensions of eating pleasure [163–168]. It is interesting to note that several similarities exist between this broader way to conceptualize eating pleasure and key dimensions we gathered in a study we recently conducted. In our focus groups [42], participants mostly defined eating pleasure in terms of taste, aesthetics, cooking, sharing a meal, relaxing, variety, nutritional aspects of food, and discovering new foods. In sum, our conceptual framework of eating pleasure provides evidence-based key dimensions and terminology that researchers may use to build consistency in the field and develop our understanding of the role of eating pleasure in the promotion of healthy eating habits. These results also highlight that *eating pleasure* may have different meanings, and that it is important for policymakers to clarify what is meant by this term when using it in policies and promotional materials.

The purpose of this scoping review was to assess whether and how eating pleasure can be a lever for healthy dietary behaviors and health. Although results were not uniform across studies, our scoping review highlights that eating pleasure is mostly (57.1%) associated with positive dietary behavior outcomes. This was especially the case for food choices, as six out of eight studies showed favorable links. However, most studies investigating the links between eating pleasure and dietary behavior outcomes were cross-sectional, hindering the assumption of causality between eating pleasure and better dietary behaviors. Nevertheless, when we looked only at intervention studies, we observed even more consistent results. In addition to leading to better diet quality [140,160], healthier portion sizes [32,37,147,148,159] and better food choices [7,160], pleasure-related strategies used in these intervention studies also increased perception of tastiness and the liking of healthy food [147], time for and perceived ease of cooking and eating [32,136,160], proportion of meals prepared at home [32] and interest in improving cooking skills [32], while decreasing disinhibition (i.e., the tendency to overeat in response to cognitive or emotional cues) [31] and food neophobia and fussiness [136]. However, although these intervention studies had promising results concerning the potential impact of using eating pleasure in the promotion of healthy eating, these findings still need to be confirmed since to date, only 11 independent intervention studies have been conducted, and these used different methodologies, evaluated different outcomes and most of the time integrated only one or a few dimensions of eating pleasure. This may be explained by the fact that the integration of eating pleasure in the promotion of healthy eating is still at a nascent stage, all intervention studies being published after 2007.

The link between eating pleasure and health outcomes has been less studied and seems more inconsistent. Only two outcomes were studied in more than two studies: body weight/ BMI and depressive symptoms. For body weight and BMI, most studies reported that eating pleasure was associated with a higher weight/BMI. However, it is important to note that the two intervention studies that documented weight/BMI observed a neutral effect. Three cross-sectional studies also assessed the link between eating pleasure and depressive symptoms, with 2 out of 3 studies showing no association between these two variables. This literature review therefore highlights a gap in research on links between eating pleasure and health-related variables, including body weight but also other health indicators less studied such as depression and anxiety symptoms, well-being, self-esteem and body weight dissatisfaction.

Tools used to measure eating pleasure in studies examining the links between eating pleasure and dietary and health outcomes used a great variety of approaches. In total, 37 different tools were used to measure eating pleasure, which greatly compromises comparisons across studies. As there seems to be no universal definition of eating pleasure in the literature, authors could define this concept according to their own perception. Inconsistent approaches and measurement of eating pleasure between studies may be at least partly responsible of divergence in results reported. The development of standardized and validated measurement tools is thus essential to have a better understanding of the links relating eating pleasure, and its key dimensions, with dietary and health outcomes.

To overcome the inconsistency in the measurement of eating pleasure, we decided to describe the associations between eating pleasure and dietary and health outcomes, taking into consideration the different dimensions that defined eating pleasure in each study. On the one hand, when eating pleasure was conceptualized as sensory experiences, mindful eating, memories, social experiences, food preparation process and food anticipation, favorable dietary outcomes were reported in the literature. In addition, findings from intervention studies suggested that sensory-based interventions (e.g., food-tasting activities; multisensory imagery), mindful eating strategies focusing on the sensory experience of eating, the rehearsal of enjoyable eating memories related to healthy foods, cooking lessons/experiences and sharing

activities may result in beneficial effects on dietary behaviors. It is worth noting that these findings are consistent with those of other studies that did not specifically focus on eating pleasure, but that observed favorable outcomes using similar intervention strategies as those identified as promising in this review [169–172]. On the other hand, visceral eating pleasure, defined as eating pleasure associated with short-term visceral impulses triggered by hunger, external cues or internal emotional urges, was more strongly related to detrimental effects on dietary outcomes according to cross-sectional studies. For health outcomes, most cross-sectional studies found that sensory experiences and novelty were favorably associated with these outcomes. Taken together, these results suggest that the different dimensions of eating pleasure are characterized by a specific pattern of associations with dietary and health outcomes, some being more favorable than others. These promising avenues may then inform whether and how the pleasure of eating can be a lever for healthy eating. However, even if this review was an essential first step to propose impactful strategies in the promotion of healthy eating, it will be essential for health professionals and policymakers to tailor messages and intervention strategies to fit the local context, including social, cultural and socioeconomic factors, to ensure that these strategies are well aligned with community needs and interests. Additional steps, including a comprehensive needs assessment, are therefore needed before using these promising strategies in policies and dietary interventions within a specific population.

It is important to highlight some gaps identified by this review. First, considering the high number of dimensions of eating pleasure (89 sub-dimensions grouped into 22 key dimensions), some of these have been less studied (e.g., place and atmosphere where food is consumed, eating on special occasions, having the choice, taking time). The lack of studies, however, does not necessarily indicate that these dimensions of eating pleasure are not favorably or unfavorably related to dietary behaviors and health outcomes. Rather, these findings highlight the fact that research on the pleasure of eating in the context of promoting healthy eating is still in its infancy and further studies are needed to get a more comprehensive perspective on the place of eating pleasure, and its multiple dimensions, in the promotion of healthy eating. In addition, some limitations of the included intervention studies need to be highlighted. First, intervention studies were more likely to include only one dimension of eating pleasure, to be short-term single exposure interventions, and to be performed with women. However, some reasons may explain these methodological choices. Short-term exposure can be more easily used to test hypotheses at the outset, allowing to identify more efficiently dimensions of eating pleasure that seem to be more promising in the promotion of healthy eating. These promising dimensions can then be tested in larger, longer-term interventions. Also, the testing of one dimension at a time allows for a better assessment of the impact of this dimension on the variables of interest, which can be very useful when designing more complex interventions afterwards. The fact that more women have been included in previous intervention studies is congruent with what is observed in other systematic reviews in nutrition, highlighting most of the time a greater proportion of women than men in these studies [173–176]. These gender disparities may be explained by gender-specific differences in many areas of nutrition, including the fact that women have a greater interest in nutrition and seek nutrition counselling more frequently than men do [177]. Finally, no intervention study examined the effectiveness of intervention strategies by sex, age or BMI, and only two comprehensive intervention studies using different dimensions of eating pleasure were reported in the present review [31,32]. Therefore, these findings highlight some research gaps and the need for conducting well-developed and evidence-based intervention studies before reaching a firm conclusion about the place of eating pleasure, and its multiple dimensions, in interventions aiming to promote healthy eating.

### Methodological strengths and limitations

This scoping review has many methodological strengths. First, the search strategy was developed with the help of a librarian specializing in nutrition-related research, limiting the risk of missing relevant documents. The search strategy included a wide range of databases, allowing us to capture a comprehensive sample from the many disciplines related to eating pleasure. In addition, the search strategy included both scientific and grey literature, allowing for a more comprehensive approach of eating pleasure. A rigorous methodology comprising two independent coders (one specialized in nutrition and one specialized in social sciences) who reviewed evidence at each step of the scoping review resulted in a substantial reduction in possible errors. In addition, the use of the scoping review *per se* is a strength, allowing us to address a broad, multidisciplinary topic using a wide range of research designs. Finally, we used the PRISMA extension for scoping reviews checklist to report these results, ensuring methodological and reporting quality [178]. Some limitations of the present review must also be acknowledged. First, we included only studies conducted in developed countries, with participants aged 5 years old and older, limiting the generalization of our results. Second, we reviewed only documents written in English or French, thus we cannot exclude the possibility that we missed some studies. Finally, several documents mentioned dimensions of eating pleasure only very briefly, thus making interpretation difficult for coders. In order not to bias the results of this review, the coders therefore analyzed them as they were described, without over-interpreting them. Accordingly, we do not exclude the possibility that some dimensions can be combined in further analyses.

### Conclusion

This comprehensive review of the literature about eating pleasure showed that this concept involves numerous key dimensions. It showed favorable links between eating pleasure and dietary outcomes and identified promising strategies for using eating pleasure in intervention strategies. However, the role occupied by studies evaluating the effects of eating pleasure on healthy eating is still minimal, and additional well-developed and evidence-based intervention studies are needed before reaching a firm conclusion about the place that eating pleasure, and its multiple dimensions, should occupy in the promotion of healthy eating.

### Supporting information

**S1 Table. Search strategies (other than Medline).**
(DOCX)

**S2 Table. Characteristics of included documents.**
(DOCX)

**S3 Table. Description of disciplines.**
(DOCX)

**S4 Table. Key dimensions of eating pleasure, with description of sub-dimensions.**
(DOCX)

**S1 File. Preferred Reporting Items for Systematic reviews and Meta-Analyses extension for Scoping Reviews (PRISMA-ScR) checklist.**
(DOCX)

### Acknowledgments

The authors are grateful to Mylène Turcotte and Annie Lapointe for their valuable assistance in the scoping review process. They also thank the specialized librarian, Daniela Zavala Mora,

for her help in developing the comprehensive search strategy, and Louisa Blair for copyediting the article.

## Author Contributions

**Conceptualization:** Alexandra Bédard, Véronique Provencher, Sophie Desroches, Simone Lemieux.

**Data curation:** Alexandra Bédard.

**Formal analysis:** Alexandra Bédard, Pierre-Olivier Lamarche, Lucie-Maude Grégoire, Catherine Trudel-Guy.

**Funding acquisition:** Simone Lemieux.

**Investigation:** Alexandra Bédard, Pierre-Olivier Lamarche, Lucie-Maude Grégoire, Catherine Trudel-Guy.

**Methodology:** Alexandra Bédard, Véronique Provencher, Sophie Desroches, Simone Lemieux.

**Project administration:** Alexandra Bédard.

**Supervision:** Simone Lemieux.

**Writing – original draft:** Alexandra Bédard, Simone Lemieux.

**Writing – review & editing:** Alexandra Bédard, Pierre-Olivier Lamarche, Lucie-Maude Grégoire, Catherine Trudel-Guy, Véronique Provencher, Sophie Desroches, Simone Lemieux.

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
