## [Decision Letter · Decision Letter 0]

13 Oct 2020

PONE-D-20-19765

Can eating pleasure be a lever for healthy eating? A systematic scoping review of eating pleasure and its links with dietary behaviors and health

PLOS ONE

Dear Dr. Bédard,

Thank you for submitting your manuscript to PLOS ONE. After careful consideration, we feel that it has merit but does not fully meet PLOS ONE’s publication criteria as it currently stands. Therefore, we invite you to submit a revised version of the manuscript that addresses the points raised during the review process.

We look forward to receiving your revised manuscript.

Kind regards,

Hans De Steur

Academic Editor

PLOS ONE

Journal Requirements:

"This work was supported by the Canadian Institutes of Health Research (author who received this grant: SL; grant number FHG 129921; URL: https://cihr-irsc.gc.ca/). The CIHR had no role in study design, data collection and analysis, decision to publish, or preparation of the manuscript. ".

i) Please provide an amended statement that declares *all* the funding or sources of support (whether external or internal to your organization) received during this study, as detailed online in our guide for authors at http://journals.plos.org/plosone/s/submit-now.  Please also include the statement “There was no additional external funding received for this study.” in your updated Funding Statement.

ii) Please include your amended Funding Statement within your cover letter. We will change the online submission form on your behalf.

Additional Editor Comments (if provided):

Dear authors,

As you will see, both reviewers have a number of comments that still need to be addressed.

Sincerely,

Reviewers' comments:

Reviewer's Responses to Questions

**Comments to the Author**

1. Is the manuscript technically sound, and do the data support the conclusions?

Reviewer #1: Yes

Reviewer #2: Yes

2. Has the statistical analysis been performed appropriately and rigorously? 

Reviewer #1: Yes

Reviewer #2: N/A

3. Have the authors made all data underlying the findings in their manuscript fully available?

Reviewer #1: Yes

Reviewer #2: No

4. Is the manuscript presented in an intelligible fashion and written in standard English?

Reviewer #1: Yes

Reviewer #2: Yes

5. Review Comments to the Author

Reviewer #1: This scoping review aiming at 1) identifying the key dimensions of eating pleasure from the existing literature, 2) summarizing currently available data on its association with dietary behaviours and health outcomes, and 3) analysing the efficiency of intervention strategies using eating pleasure to promote healthy eating is of great interest for the field of behavioural nutrition. This an ambitious and well-conducted piece of research that clearly addresses major gaps in the literature.

Without a clear and shared definition of eating pleasure, various concepts have been associated with eating pleasure and the effort of summarizing these concepts and their measures is very useful for future research. The analysis of the quite mixed evidence regarding the association between eating pleasure and dietary behaviours/health outcomes among cross-sectional and interventional studies considering the different dimensions of eating pleasure is greatly informative and could inform interventions design in the future.

Some comments and suggestions below.

Introduction

L27-32 Why framing this research in Canada? Probably better to aim more global as you included studies from various developed countries in the scoping review (Table 3).

L33 The assumption “Current strategies used in the promotion of healthy eating are ineffective for most of the population” is a bit of a stretch and a stronger background would be needed. Longitudinal studies are a start, but it is still hard to know whether the nutritional quality of the diet would not have been even worse without any promotion of healthy eating.

L57 Petit et al. is not the most appropriate example to cite as brain activity is hard to link with dietary patterns/nutritional quality.

Methods and results sections are well written, and the data are clearly and efficiently summarized.

Discussion

The identification of the 22 key dimensions is a very important part of this work, but the high number of dimensions will make hard to apply these findings in future research. It is also highlighted that policymakers need to clarify what is meant by eating pleasure when it is used in dietary guidelines (L715-716). In my opinion, it could be the role of researchers to provide a clear framework to inform future public policies, i.e. stating more clearly what pleasure should mean in policies. The paragraph from L753 is of great importance as it highlights that only a certain conceptualisation of eating pleasure may have a favourable effect on dietary behaviours and health outcomes. I would recommend making more of these findings in the discussion, e.g. with a summary table/figure that groups together the dimensions based on their likely favourable or unfavourable impact and ordering the dimensions based on level/strength of evidence. This table/figure could also refer to the measures/intervention tools previously used for the key dimensions. As stated L770 “Results of this review highlight the need for well-developed and evidence-based intervention studies before reaching a firm conclusion about the place of eating pleasure, and its multiple dimensions, in interventions aiming to promote healthy eating” and a table/figure summarizing how to do so would be very useful for future research/policies and would enhance the impact of this paper.

Reviewer #2: Introduction

L43: some papers have suggested that the link tasty = unhealthy might be related to cultural differences. It would be good to include this.

L47 - 74: A whole page is foreseen to discuss a positive link between eating pleasure and diet quality while only one sentence is used to mention that several other authors found a negative link. I would recommend that the authors revise this part and make it more balance, with a less or more equal introduction of the positive and negative links.

L98: a reference about scoping review might be appropriate and maybe also a small definition especially given that it is later compared with the better known systematic review

Materials

L108: what was adapted and why compared to the Arksey and O'Malley methodological framework?

L187: in order to lower the barrier for other researchers, I would suggest that all search strategies are enclosed and not just are available upon request. They can always be added as supplementary material if the authors fear that he article might be too long when the search strategies would be included in the main manuscript.

Results

How was the discipline of the first author determined?

Why did you not specify which disciplines fell under 'multiply disciplines'?

Wouldn't it be interesting to also provide more information about the discipline of the principal investigator (last author)? That could also give some additional insights.

L582: be carefull when working with percentages for Q3, as it only comprises 11 studies.

Discussion

L719: " highlights that eating pleasure is mostly associated with positive dietary behavior outcomes." => could you back this with the % of studies?

L740: ..., suggesting mostly no association between these two variables. => why mostly? Is that based upon 2 studies out of 3? Be careful with making a clear statement when only considering 3 studies.

L751: any suggestions of tools which might be seen as more standardized and validated for the measuring eating pleasure?

L766: Intervention studies were more likely to include only one dimension of eating pleasure, to be short-term single exposure interventions, and to be performed with women. => could you short elaborate on the reasons why this might be the case? For instance, why would they not work with women, focus on short-term single exposure,...? Is this related to the lack of a standardized and validated measure for eating pleasure? I believe that adding some insights about the reasons of these limitations might help people to tackle the limitations and help (young) scientists when setting up new intervention experiments.

Reference

=> reference 6 begins with a dot, please correct

6. PLOS authors have the option to publish the peer review history of their article (what does this mean?). If published, this will include your full peer review and any attached files.

Reviewer #1: No

Reviewer #2: No

---

## [Author Response · Author response to Decision Letter 0]

11 Nov 2020

Response to reviewers

We would like to thank the Editor and reviewers for their insightful comments and suggestions. You will find below our point-by-point response to each of these comments/suggestions. Please note that page and line numbers refer to those of the revised manuscript with track changes.

EDITOR – Journal Requirements

We carefully reviewed our article and the attached documents to ensure that they comply with PLOS ONE’s style requirements.

2. Please provide an amended statement that declares *all* the funding or sources of support (whether external or internal to your organization) received during this study, as detailed online in our guide for authors. Please also include the statement “There was no additional external funding received for this study.” in your updated Funding Statement.

As reported in our initial submission, we have received funding only from the Canadian Institutes of Health Research (CIHR) for this work. There was no additional external funding received for this study, and we now include this statement in our updated Funding Statement within our cover letter. 

3. Have the authors made all data underlying the findings in their manuscript fully available? Reviewer #2’s answer: No

The PLOS ONE Data policy requires authors to make all data and related metadata underlying the findings fully available without restriction. Data should be deposited in an appropriate public repository, unless already provided as part of the submitted article. Accordingly, we would like to point out that we comply with the PLOS ONE policy, as all relevant data are within our manuscript and its supporting information files.

REVIEWER 1: Review comments

1. L27-32 Why framing this research in Canada? Probably better to aim more global as you included studies from various developed countries in the scoping review (Table 3).

In the initial submission, we reported eating habits of our population (i.e., Canadians) only as an example to illustrate suboptimal diet quality in developed countries. However, we agree that the results of this literature review cover several developed countries and that these countries all share similar suboptimal dietary habits, i.e., consuming not enough fruits, vegetables and wholes grains, and too much processed foods high in energy, sodium, free sugars and saturated fats. Accordingly, and as suggested, we now frame this research more globally in this paragraph. (Introduction: p. 3 lines 43-47).

2. L33 The assumption “Current strategies used in the promotion of healthy eating are ineffective for most of the population” is a bit of a stretch and a stronger background would be needed. Longitudinal studies are a start, but it is still hard to know whether the nutritional quality of the diet would not have been even worse without any promotion of healthy eating.

We agree that this assumption is a bit exaggerated. In fact, our intention is not to underline that current strategies are not effective, but rather to emphasize the necessity to identify food policies and interventions that may have a greater impact on individual’s diet quality over time. We have clarified this point in the revised version of the manuscript (Introduction: p. 3 lines 48-63).

3. L57 Petit et al. is not the most appropriate example to cite as brain activity is hard to link with dietary patterns/nutritional quality.

We agree with the reviewer that brain activity may be hard to link with nutritional quality. However, in addition to brain activity, Petit et al. reported in their paper that focusing on the tastiness of food also increased the percentage of healthy food choices compared with a control condition (Introduction: p. 4-5 lines 88-89). In that context, we still believe that the study of Petit et al. is an adequate example of studies investigating the link between eating pleasure and diet quality/dietary habits.

4. The identification of the 22 key dimensions is a very important part of this work, but the high number of dimensions will make hard to apply these findings in future research. It is also highlighted that policymakers need to clarify what is meant by eating pleasure when it is used in dietary guidelines (L715-716). In my opinion, it could be the role of researchers to provide a clear framework to inform future public policies, i.e. stating more clearly what pleasure should mean in policies. The paragraph from L753 is of great importance as it highlights that only a certain conceptualisation of eating pleasure may have a favourable effect on dietary behaviours and health outcomes. I would recommend making more of these findings in the discussion, e.g. with a summary table/figure that groups together the dimensions based on their likely favourable or unfavourable impact and ordering the dimensions based on level/strength of evidence. This table/figure could also refer to the measures/intervention tools previously used for the key dimensions. As stated L770 “Results of this review highlight the need for well-developed and evidence-based intervention studies before reaching a firm conclusion about the place of eating pleasure, and its multiple dimensions, in interventions aiming to promote healthy eating” and a table/figure summarizing how to do so would be very useful for future research/policies and would enhance the impact of this paper.

We agree with the reviewer about the relevance to add a summary table of our findings in our revised manuscript to recap the actual evidence about the place of eating pleasure in the promotion of healthy eating. Accordingly, a summary table has been added in the results section (see Table 7 in the revised manuscript). As suggested, this table groups together the dimensions based on their likely favorable or unfavorable impact and presents the dimensions in an order based on the level of evidence. Also, where applicable, we indicated in this table the intervention strategies that have shown promising effects for each dimension. We agree with the reviewer that this summary table will greatly enhance the impact of the findings, suggesting ways to use eating pleasure in future research/policies. We have adjusted the results and discussion sections as a result of this addition (Results: p. 54 lines 731-734; Discussion: p. 59 lines 816-837).

Also, we agree with the reviewer that the high number of dimensions (89 sub-dimensions grouped into 22 key dimensions) of eating pleasure could make hard to apply these findings in future research. In this regard, the new summary table will be very informative. However, it is important to keep in mind that some dimensions have been less studied (e.g., place and atmosphere where food is consumed, eating on special occasions, having the choice, taking time). The lack of studies, however, does not necessarily indicate that these dimensions of eating pleasure are not favorably or unfavorably related to dietary behaviors and health outcomes. Rather, these findings highlight the fact that research on the pleasure of eating in the context of promoting healthy eating is still in its infancy and further studies are needed to get a more comprehensive perspective on the place of eating pleasure, and its multiple dimensions, in the promotion of healthy eating.We made sure to underline this point in the discussion section (Discussion: p. 61 lines 848-856). 

Finally, this review is an essential first step to assess whether and how eating pleasure can be a lever for healthy dietary behaviors and health. However, we believe that it is probably too soon to state what pleasure should mean in policies. To propose impactful strategies in the promotion of healthy eating, it will be essential for health professionals and policymakers to tailor messages and intervention strategies to fit the local context, including social, cultural and socioeconomic factors, to ensure that these strategies are well aligned with community needs and interests. Additional steps, including a comprehensive needs assessment, are therefore needed before using these promising strategies in policies and dietary interventions within a specific population. Therefore, instead of stating what pleasure should mean in policies, we made sure to highlight in the results section the dimensions that appear to be more favorably related to healthy eating (see Table 7), and to state in the discussion section the next step that health professionals and policymakers should take (e.g., needs assessment) before integrating eating pleasure in the promotion of healthy eating (Discussion: p. 59-60 lines 837-844). 

REVIEWER 2 : Review comments

1. L43: some papers have suggested that the link tasty = unhealthy might be related to cultural differences. It would be good to include this.

As suggested, we now underline in the introduction section that culture may influence this perception (Introduction: p. 4 lines 68-72).

2. L47 - 74: A whole page is foreseen to discuss a positive link between eating pleasure and diet quality while only one sentence is used to mention that several other authors found a negative link. I would recommend that the authors revise this part and make it more balance, with a less or more equal introduction of the positive and negative links.

As suggested, we now introduce in more detail the unfavorable links between eating pleasure and dietary/health outcomes found in some previous studies (Introduction: p. 5 lines 100-108).

3. L98: a reference about scoping review might be appropriate and maybe also a small definition especially given that it is later compared with the better known systematic review.

As suggested, we now refer to some key methodological papers in this paragraph, and we have added a small definition (Introduction: p. 7 lines 138-144).

4. L108: what was adapted and why compared to the Arksey and O'Malley methodological framework?

Arksey and O’Malley published one of the first methodological frameworks for conducting scoping reviews in 2005. While this framework provided an excellent methodological foundation, Levac et al. outlined in 2010 the lack of sufficient methodological description or detail about the data analysis process in published scoping reviews, making it challenging for readers to understand how study findings were determined. Therefore, they proposed recommendations that clarify and enhance each stage of the framework. Recommendations include: clarifying and linking the purpose and research question (stage one); balancing feasibility with breadth and comprehensiveness of the scoping process (stage two); using an iterative team approach to selecting studies (stage three) and extracting data (stage four); incorporating a numerical summary and qualitative thematic analysis, reporting results, and considering the implications of study findings to policy, practice, or research (stage five); and incorporating consultation with stakeholders as a required knowledge translation component of scoping study methodology (stage six). We now give a little more detail surroundings the methodological framework used in the present review, highlighting that our methodology was based on the Arksey and O’Malley methodological framework for scoping reviews and enhanced according to the recommendations of Levac et al. aimed at clarifying each stage of the framework (Methodology: p. 7 lines 151-153).

5. L187: in order to lower the barrier for other researchers, I would suggest that all search strategies are enclosed and not just are available upon request. They can always be added as supplementary material if the authors fear that the article might be too long when the search strategies would be included in the main manuscript.

As suggested, all search strategies are now enclosed, either directly in the manuscript (Medline) or in supporting information files (all other search strategies; see S1 Table).

6. How was the discipline of the first author determined? Why did you not specify which disciplines fell under 'multiply disciplines'? Wouldn't it be interesting to also provide more information about the discipline of the principal investigator (last author)? That could also give some additional insights.

As stated in the legend of Table 3, we used the first author’s affiliation to determine the first author’s discipline. We now also state this point in the methodological section (see Table 2). In addition, in the legend of Table 3, we now specify which disciplines fell under “multiple fields”. Finally, we chose not to present the discipline of the principal investigator since, for most of the studies, the first author’s discipline was the same as the one of the principal investigator. 

7. L582: be carefull when working with percentages for Q3, as it only comprises 11 studies.

In order to take this comment into account, we made sure that each percentage presented in the text was preceded by the number of papers related to that result.

8. L719: " highlights that eating pleasure is mostly associated with positive dietary behavior outcomes." => could you back this with the % of studies?

As suggested, we have added the percentage of studies reporting favorable dietary outcomes (i.e., 57.1%; Discussion: p. 57 line 780).

9. L740: ..., suggesting mostly no association between these two variables. => why mostly? Is that based upon 2 studies out of 3? Be careful with making a clear statement when only considering 3 studies.

We are more cautious when reporting this result in the discussion section. We now state that “Three cross-sectional studies also assessed the link between eating pleasure and depressive symptoms, with 2 out of 3 studies showing no association between these two variables.” (Discussion: p. 58 lines 802-804).

10. L751: any suggestions of tools which might be seen as more standardized and validated for the measuring eating pleasure?

Our literature review revealed that 37 different tools were used to measure eating pleasure in studies examining the links between eating pleasure and dietary and health outcomes. Among them, only a few tools were validated, and no tool comprehensively measured the pleasure of eating. However, it is important to underline that our literature review did not aim to list all the tools that measure eating pleasure. Therefore, we find it premature to propose specific tools that should be used in future studies. However, we agree with the reviewer that this research question is important and should be the purpose of a next scoping review.

11. L766: Intervention studies were more likely to include only one dimension of eating pleasure, to be short-term single exposure interventions, and to be performed with women. => could you short elaborate on the reasons why this might be the case? For instance, why would they not work with women, focus on short-term single exposure,...? Is this related to the lack of a standardized and validated measure for eating pleasure? I believe that adding some insights about the reasons of these limitations might help people to tackle the limitations and help (young) scientists when setting up new intervention experiments.

We agree with the reviewer that adding some insights about the reasons of these limitations could be of great relevance. Accordingly, we are now proposing some reasons that may explain these methodological choices in the discussion section (Discussion: p. 61 lines 859-870). First, research on the pleasure of eating in the context of promoting healthy eating is still in its infancy. Short-term exposure can be more easily used to test hypotheses at the outset, allowing to identify more efficiently dimensions of eating pleasure that seem to be more promising in the promotion of healthy eating. These promising dimensions can then be tested in larger, longer-term interventions. Also, the testing of one dimension at a time allows for a better assessment of the impact of this dimension on the variables of interest, which can be very useful when designing more complex interventions afterwards. The fact that more women have been included in previous intervention studies is congruent with what is observed in other systematic reviews in nutrition, highlighting most of the time a greater proportion of women than men in these studies. These gender disparities may be explained by gender-specific differences in many areas of nutrition, including the fact that women have a greater interest in nutrition and seek nutrition counselling more frequently than men do.

12. Reference 6 begins with a dot, please correct.

We have made this change.

---

## [Decision Letter · Decision Letter 1]

8 Dec 2020

Can eating pleasure be a lever for healthy eating? A systematic scoping review of eating pleasure and its links with dietary behaviors and health

PONE-D-20-19765R1

Dear Dr. Bédard,

We’re pleased to inform you that your manuscript has been judged scientifically suitable for publication and will be formally accepted for publication once it meets all outstanding technical requirements.

Kind regards,

Hans De Steur

Academic Editor

PLOS ONE

Additional Editor Comments (optional):

Reviewers' comments:

Reviewer's Responses to Questions

**Comments to the Author**

1. If the authors have adequately addressed your comments raised in a previous round of review and you feel that this manuscript is now acceptable for publication, you may indicate that here to bypass the “Comments to the Author” section, enter your conflict of interest statement in the “Confidential to Editor” section, and submit your "Accept" recommendation.

Reviewer #1: All comments have been addressed

Reviewer #2: All comments have been addressed

2. Is the manuscript technically sound, and do the data support the conclusions?

Reviewer #1: Yes

Reviewer #2: Yes

3. Has the statistical analysis been performed appropriately and rigorously? 

Reviewer #1: Yes

Reviewer #2: N/A

4. Have the authors made all data underlying the findings in their manuscript fully available?

Reviewer #1: Yes

Reviewer #2: Yes

5. Is the manuscript presented in an intelligible fashion and written in standard English?

Reviewer #1: Yes

Reviewer #2: Yes

6. Review Comments to the Author

Reviewer #1: Thank you for an enhanced manuscript. Table 7 provides a useful summary of the results and the discussion now states more clearly what are the next steps for both the research community and policy makers.

Reviewer #2: (No Response)

7. PLOS authors have the option to publish the peer review history of their article (what does this mean?). If published, this will include your full peer review and any attached files.

Reviewer #1: No

Reviewer #2: No

---

## [Editor Report · Acceptance letter]

10 Dec 2020

PONE-D-20-19765R1 

Can eating pleasure be a lever for healthy eating? A systematic scoping review of eating pleasure and its links with dietary behaviors and health 

Dear Dr. Bédard:

I'm pleased to inform you that your manuscript has been deemed suitable for publication in PLOS ONE. Congratulations! Your manuscript is now with our production department. 

Kind regards, 

on behalf of

Dr. Hans De Steur 

Academic Editor

PLOS ONE